# ARE TIME-SERIES FOUNDATION MODELS DEPLOYMENT-READY? A SYSTEMATIC STUDY OF ADVERSARIAL ROBUSTNESS ACROSS DOMAINS

## ABSTRACT

Time-Series Foundation Models (TSFMs) are rapidly transitioning from research prototypes to core components of critical decision-making systems, driven by their impressive zero-shot forecasting capabilities. However, as their deployment surges, a critical blind spot remains: their fragility under adversarial attacks. This lack of scrutiny poses severe risks, particularly as TSFMs enter high-stakes environments vulnerable to manipulation. We present a systematic, diagnostic study arguing that for TSFMs, robustness is not merely a secondary metric but a prerequisite for trustworthy deployment comparable to accuracy. Our evaluation framework, explicitly tailored to the unique constraints of time series, incorporates normalized, sparsity-aware perturbation budgets and unified scale-invariant metrics across white-box and black-box settings. Across six representative TSFMs, we demonstrate that current architectures are alarmingly brittle: even small perturbations can reliably steer forecasts toward specific failure modes, such as trend flips and malicious drifts. We uncover TSFM-specific vulnerability patterns, including horizon-proximal brittleness, increased susceptibility with longer context windows, and weak cross-model transfer that points to model-specific failure modes rather than generic distortions. Finally, we show that simple adversarial fine-tuning offers a cost-effective path to substantial robustness gains, even with out-of-domain data. This work bridges the gap between TSFM capabilities and safety constraints, offering essential guidance for hardening the next generation of forecasting systems.[1]

## 1 INTRODUCTION

Time-series forecasting serves as the backbone of critical decision-making in finance, energy, transportation, and healthcare (Tang et al., 2022; Mystakidis et al., 2024; Profillidis & Botzoris, 2018; Morid et al., 2023). Driven by the success of foundation models in vision (Kirillov et al., 2023; Brooks et al., 2024; Rombach et al., 2022) and language (Dubey et al., 2024; Liu et al., 2024a; Brown et al., 2020), a new generation of Time-Series Foundation Models (TSFMs) (Liang et al., 2024) has emerged. Pretrained on massive cross-domain datasets, these models enable zero-shot forecasting in dynamic, data-scarce environments where traditional supervised models often struggle. However, as TSFMs transition from research prototypes to deployment in high-stakes systems, a fundamental question remains unanswered: *Are these models robust to adversarial manipulation?*

This question is urgent. Unlike images or text, time-series data lacks human-perceptible semantic structure, making subtle manipulations difficult to detect yet capable of triggering cascading failures, from automated trading losses to grid instability. While adversarial robustness is well-documented in vision (Szegedy et al., 2013; Goodfellow et al., 2014; Heinrich et al., 2020; Costa et al., 2024; Madry et al., 2017; Chen et al., 2017; Guo et al., 2019; Moosavi-Dezfooli et al., 2016) and language (Shayegani et al., 2023; He & Vechev, 2023; Dong et al., 2021; Wang et al., 2021a;b; Koulakos et al., 2024), the security landscape for TSFMs remains dangerously underexplored. Although recent studies have probed Large Language Model (LLM)–based forecasters (Liu et al., 2025; Liu & Jiang, 2025), native non-LLM TSFMs rely on fundamentally different architectural inductive biases. It is therefore unclear whether they inherit the vulnerabilities of their LLM counterparts or exhibit distinct

---

[1]The code is available at `https://anonymous.4open.science/r/Attack_TSFMs-9622/`

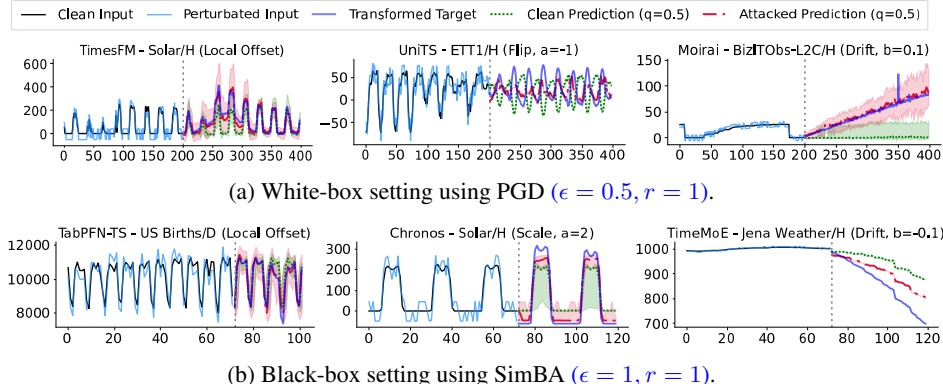

(a) White-box setting using PGD ($\epsilon = 0.5, r = 1$).

(b) Black-box setting using SimBA ($\epsilon = 1, r = 1$).

Figure 1: **Visualization of targeted adversarial attacks on TSFMs across diverse domains.** This figure illustrates how adversarial perturbations guide model forecasts toward specific target behaviors (i.e., transformed target). All perturbations use a variance-normalized budget, where $\epsilon$ is the per-step bound after normalization and $r$ is the fraction of perturbed time steps. The $q$ indicates prediction quantile (e.g., $q = 0.5$ for median, $q = 0.1/0.9$ for uncertainty bands).

failure modes. Given the rapid adoption of these models, a unified, time-series–grounded robustness evaluation is a critical prerequisite for their safe deployment.

In this work, we adopt a *diagnostic* rather than purely algorithmic perspective to conduct a systematic robustness study of widely used non-LLM TSFMs. We introduce an evaluation framework tailored to the unique constraints of temporal data, utilizing variance-normalized perturbation budgets and sparse constraints that reflect realistic threat models. To ensure fair comparison across heterogeneous datasets, we propose a unified, scale-invariant metric that consistently evaluates both untargeted (degradation) and targeted (goal-seeking) attacks. As illustrated in Figure 1, our diagnostics reveal that *even small, imperceptible perturbations can reliably steer TSFM forecasts toward attacker-specified behaviors*, such as inducing malicious drifts, trend reversals, or amplitude shifts. Our contributions are summarized as follows:

- **A time-series–grounded evaluation framework.** We propose a comprehensive assessment protocol that systematically varies adversarial objectives, capabilities, and knowledge. By standardizing metrics and constraints, we enable rigorous, fair comparisons across diverse TSFM architectures.

- **Identification of unique vulnerability patterns.** Across six representative TSFMs, we uncover distinct failure modes not typically seen in other modalities. These include horizon-proximal brittleness, context-length–induced vulnerability amplification, and weak cross-model transfer, etc.

- **Scalable, transferable defenses.** We demonstrate that lightweight adversarial fine-tuning significantly improves worst-case robustness. Crucially, we find that these defenses are highly transferable: cross-domain latent adversarial training recovers a large fraction of in-domain gains, offering a practical pathway to secure TSFMs even when task-specific training data is unavailable.

Taken together, these findings provide a necessary reality check for the field, quantifying the risks of current architectures while offering actionable guidance for building resilient, deployment-ready forecasting systems.

## 2 RELATED WORK

**Foundation Models for Time Series Forecasting.**    Inspired by the success of foundation models in language (Brown et al., 2020), a growing number of time-series foundation models have emerged (Das et al., 2024; Rasul et al., 2023; Ansari et al., 2024; Shi et al., 2025; Woo et al., 2024; Ekambaram et al., 2024; Hoo et al., 2025; Goswami et al., 2024; Gao et al., 2024; Liu et al., 2024b; Darlow et al., 2024; Yeh et al., 2023; Zhang et al., 2025; Cao et al., 2024b; Garza et al., 2023; Cao et al., 2024a; Prabhakar Kamarthi & Prakash, 2024). Despite differences in backbone architecture, optimization objectives, and data regimes, these TSFMs share a common objective: achieving universal forecasting

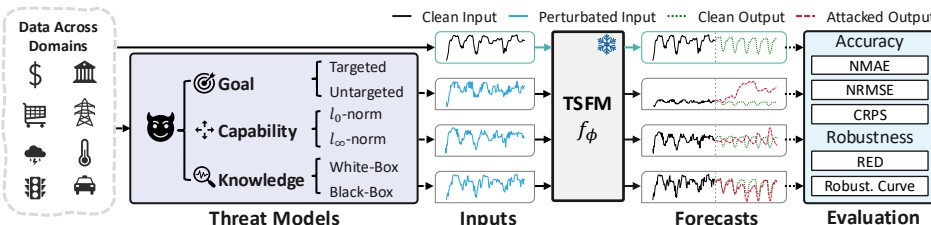

Figure 2: **Overview of the adversarial evaluation protocol for time-series foundation models.** We evaluate TSFMs across diverse domains under a unified adversarial framework. Adversarial perturbations are applied to clean inputs, which are then passed through the TSFM to produce perturbed forecasts. We assess the impact of these attacks using both accuracy and robustness metrics.

capability. However, this pursuit of universality and accuracy has often come at the expense of robustness (Ilyas et al., 2019; Su et al., 2018). As TSFMs move toward real-world deployment, their susceptibility to adversarial perturbations becomes a central reliability concern.

**Adversarial Robustness in Deep Neural Networks.** It is well established that deep neural networks are highly vulnerable to imperceptible input perturbations that can lead to significant prediction errors (Szegedy et al., 2013; Goodfellow et al., 2014). This discovery sparked a wave of research on robustness evaluation, resulting in the development of standardized benchmarking frameworks and attack methods in computer vision (Carlini et al., 2019; Liu et al., 2022; Gao et al., 2022; Dong et al., 2020; Croce et al., 2021). With the rapid advancement and widespread adoption of large language models and vision-language foundation models, adversarial robustness in these systems has also attracted growing attention (Shayegani et al., 2023; He & Vechev, 2023; Perez et al., 2022; Zhao et al., 2023; Schlarmann & Hein, 2023).

In contrast, adversarial robustness in time series models remains underexplored. Prior work has largely focused on classification tasks (Siddiqui et al., 2020; Rathore et al., 2020; Karim et al., 2020; Fawaz et al., 2019; Ding et al., 2023), or on constructing adversarial examples for specific forecasting architectures (Dang-Nhu et al., 2020; Yoon et al., 2022; Liu et al., 2023). While these studies demonstrate that time series models are indeed susceptible to adversarial perturbations, they primarily address small-scale models and narrow domain settings. Recent studies have begun examining robustness in LLM-based forecasting systems (Liu et al., 2025; Liu & Jiang, 2025). However, since these works focus on LLM-based pipelines that are distinct from time-series-grounded TSFMs, existing robustness work on LLM-based models does not directly apply to time-series-oriented designs. The robustness of large-scale, pretrained time-series-grounded TSFMs has yet to be systematically investigated, and their behavior under adversarial conditions remains unknown.

## 3 ADVERSARIAL EVALUATION FRAMEWORK AND DEFENSE METHODS

Figure 2 outlines our evaluation protocol, which comprises threat specification (Section 3.2), perturbation construction (Section 3.3), and robustness metrics (Section 3.4). We further investigate readily deployable defenses in Section 3.5.

### 3.1 PRELIMINARIES

**Time-Series Forecasting.** We consider a univariate time series $\mathbf{x}_{1:T} = \{x_\tau\}_{\tau=1}^T$, where each observation $x_\tau \in \mathbb{R}$ corresponds to the value at time step $\tau$. A forecasting model $f_\phi$ with parameters $\phi$ maps an input sequence of length $L$ to a prediction over the future $T$ steps $f_\phi : \mathbf{x}_{t-L:t} \mapsto \hat{\mathbf{x}}_{t+1:t+T}$, where $\hat{\mathbf{x}}_{t+1:t+T}$ is the predicted future trajectory. The model is typically trained by minimizing the expected forecasting loss: $\min_\phi \mathbb{E}_{\mathbf{x} \sim \mathcal{P}(\mathcal{D}),\, t \sim \mathcal{P}(\mathcal{T})} [\mathcal{L}(f_\phi(\mathbf{x}_{t-L:t}), \mathbf{x}_{t+1:t+T})]$, where $\mathcal{P}(\mathcal{D})$ is the data distribution, $\mathcal{P}(\mathcal{T})$ is the sampling distribution over timestamps, and $\mathcal{L}$ is the loss function. TSFMs are typically pretrained on large-scale, multi-domain datasets, and their parameters $\phi$ remain frozen during downstream deployment.

Table 1: Parameter settings for different target transformations used in targeted adversarial attacks. Each transformation modifies the forecast in a specific pattern by configuring the scaling factor $a$ and time-dependent bias $c(\tau)$. Here, $b \in \mathbb{R}$ controls the drift strength, $\delta_\tau \in \mathbb{R}$ specifies localized shifts, and $\mathcal{I} \subseteq \{1, \ldots, T\}$ denotes the perturbed forecast steps. See Figure 3 for transformation examples.

| Transformation | Parameters | Description |
|---|---|---|
| Scaling | $a > 0, c(\tau) = 0$ | Uniformly scales the forecast |
| Flipping | $a < 0, c(\tau) = 0$ | Reflects the sequence |
| Drifting | $a = 1, c(\tau) = b \cdot \tau$ | Adds global linear drift |
| Local Offsetting | $a = 1, c(\tau) = \delta_\tau$ if $\tau \in \mathcal{I}$ | Perturbs selected steps only |

**Threat Model.** An *adversarial example* is generated by adding a small perturbation $\boldsymbol{\delta} \in \mathbb{R}^L$ to the input, producing $\mathbf{x}^{\text{adv}} = \mathbf{x} + \boldsymbol{\delta}$, such that the model's output deviates significantly from that on the clean input $\mathbf{x}$. The *threat model* defines the adversary's assumptions along three dimensions: *goal*, *capability*, and *knowledge* (Carlini et al., 2019). The goal specifies the intended effect of the attack: either *untargeted*, aiming to degrade performance without a specific outcome, or *targeted*, steering predictions toward a chosen trajectory. The capability constrains the allowable perturbation, typically bounded by an $\ell_p$-norm constraint $\|\boldsymbol{\delta}\|_p \leq \epsilon$, where $\epsilon$ is the perturbation budget and $p \in \{0, 2, \infty\}$ defines the geometry. The knowledge describes the adversary's access to the model: in a *white-box* setting, the attacker has full access to architecture and parameters, while in a *black-box* setting, access is restricted to input-output queries, such as through an API.

## 3.2 THREAT MODEL SPECIFICATION FOR TSFMs

**Objective.** Given an input sequence $\mathbf{x}_{t-L:t} \in \mathbb{R}^L$, a pre-trained forecasting model $f_\phi$, and a reference sequence $\mathbf{y} \in \mathbb{R}^T$ (which may represent the ground-truth, clean prediction, or an attacker-defined target), the goal is to find a perturbation $\boldsymbol{\delta} \in \mathcal{S} \subseteq \mathbb{R}^L$ such that the model's output on the perturbed input significantly deviates from—or closely matches—the reference. Specifically, we define the general attack objective as:

$$\boldsymbol{\delta}^* = \arg \max_{\boldsymbol{\delta} \in \mathcal{S}} \left[ \sigma \cdot \mathcal{L}(f_\phi(\mathbf{x}_{t-L:t} + \boldsymbol{\delta}), \mathbf{y}) \right], \tag{1}$$

where $\mathcal{L}(\cdot, \cdot)$ is a forecasting loss function (e.g., MSE, MAE), and $\sigma \in \{+1, -1\}$ indicates the attack direction: $+1$ for *untargeted attacks* (maximizing prediction error with respect to the clean output), and $-1$ for *targeted attacks* (minimizing error toward a predefined target $\mathbf{y}$). The perturbation set $\mathcal{S} \subseteq \mathbb{R}^L$ specifies the allowable perturbations. We denote the overall attack objective as:

$$g_\phi(\boldsymbol{\delta}) := \sigma \cdot \mathcal{L}(f_\phi(\mathbf{x}_{t-L:t} + \boldsymbol{\delta}), \mathbf{y}), \tag{2}$$

which unifies both attack types under a single formulation. Due to the unavailability of ground-truth future values at inference time, we treat the model's clean prediction as the attack target $\mathbf{y}$.

**Building Targets for Targeted Attacks.** To evaluate the performance of TSFMs under targeted attacks, we first construct appropriate target trajectories. In practice, attackers often seek to introduce subtle but systematic deviations—such as periodic shifts or gradual drifts—that are difficult to detect yet can accumulate significant downstream effects. To support controlled experimentation, the extent of deviation from the clean forecast should be adjustable, enabling us to examine how susceptible the model is to both mild and aggressive manipulations.

To this end, we define a family of transformation functions $\mathcal{M}(\cdot)$ that generate adversarial targets by applying structured modifications to the clean forecast. Given a clean prediction $\hat{\mathbf{x}} = \{\hat{x}_\tau\}_{\tau=1}^T$, the transformed target sequence $\mathbf{y} = \{y_\tau\}_{\tau=1}^T$ is computed as:

$$y_\tau = \mathcal{M}(\hat{x}_\tau; a, c) = a \cdot \hat{x}_\tau + c(\tau), \tag{3}$$

where $a \in \mathbb{R}$ controls the amplitude (scaling), and $c(\tau)$ is a time-dependent bias function. By tuning $a$ and the shape of $c(\tau)$, we instantiate various adversarial patterns with adjustable distortion levels. Table 1 summarizes the transformation types considered in our evaluation, and Figure 1 and 3 illustrates visual examples of the resulting targets.

**Perturbation Budget.** The perturbation budget directly reflects the strength of the attack. Larger perturbations tend to cause greater deviations in model outputs, but excessive distortion may lead to unrealistic inputs that compromise both the fairness of robustness evaluation and the plausibility of real-world scenarios. For time-series data, two factors are particularly important: the number of perturbed time steps, and the magnitude of each perturbation. As a result, a single $\ell_p$-norm constraint is insufficient and finer-grained control is required. Motivated by this, we impose a *hybrid norm constraint* that jointly bounds perturbation sparsity and amplitude:

$$\mathcal{S} = \left\{ \boldsymbol{\delta} \in \mathbb{R}^L \ : \ \|\boldsymbol{\delta}\|_0 \le rL, \ \|\boldsymbol{\delta}\|_\infty \le \epsilon \right\}, \tag{4}$$

where $r \in (0, 1]$ denotes the *perturbation ratio* (the fraction of time point modified), and $\epsilon > 0$ is the *per-element perturbation bound* (the maximum change per time point). Based on our empirical observation that time steps closer to the forecast horizon are more vulnerable (see Section 4.2), we default to perturbing the last $rL$ time steps when $r < 1$, unless stated otherwise. To ensure comparability across datasets with different scales, we use a variance-normalized perturbation budget. Throughout the paper, the $\epsilon$ values we report are scale-free coefficients, and the actual per-step bound applied is $\epsilon^* = \epsilon \cdot \mathrm{var}(\mathbf{x})$, where $\mathrm{var}(\mathbf{x})$ denotes the variance of the input sequence.

### 3.3 Adversarial Perturbation Optimization

We construct adversarial perturbations under different access assumptions. The *white-box* setting assumes full gradient access and serves as a worst-case probe. The *black-box* setting restricts the adversary to input-output queries without knowledge of parameters, gradients, or training data, which more closely reflects practical deployment scenarios.

**White-Box Setting.** In the white-box setting, the attacker has full access to the model architecture and parameters $\phi$. We adopt *Projected Gradient Descent (PGD)* (Madry et al., 2017), a widely used and reliable attack method. PGD iteratively updates the perturbation as

$$\boldsymbol{\delta}^{k+1} = \Pi_{\mathcal{S}} \left( \boldsymbol{\delta}^k + \alpha \cdot \mathrm{sgn}\big(\nabla_{\boldsymbol{\delta}} g_\phi(\boldsymbol{\delta}^k)\big) \right), \tag{5}$$

where $\alpha$ is the step size, $\Pi_{\mathcal{S}}$ is projection onto the feasible set $\mathcal{S}$, and $k$ is the iteration index. We also report results of the *Fast Gradient Sign Method (FGSM)* (Goodfellow et al., 2015) in Appendix E.3.

**Black-Box Setting.** In the black-box setting, the attacker must estimate effective perturbations using only model queries. We implement two representative methods: *Zero-Order Optimization (ZOO)* (Chen et al., 2017) and the *Simple Black-box Attack (SimBA)* (Guo et al., 2019). *ZOO* approximates gradients via finite differences. For the $i$-th component,

$$\nabla_i g_\phi(\boldsymbol{\delta}) \approx \frac{g_\phi(\boldsymbol{\delta} + \mu \mathbf{u}_i) - g_\phi(\boldsymbol{\delta})}{2\mu}, \tag{6}$$

where $\mathbf{u}_i$ is the $i$-th basis vector and $\mu > 0$ is a small constant. We further adopt a *ZOO-Adam* variant that applies the Adam optimizer to the estimated gradients for improved stability and convergence.

*SimBA* performs query-efficient random search over an orthogonal basis. At each iteration, a direction $\mathbf{q} \in Q$ is sampled without replacement, and the perturbation is updated if it improves the attack objective. To better match the structure of time-series data, we explore three basis designs: Cartesian (point-wise), DCT (low-frequency), and Wavelet (time-frequency). These yield perturbations with distinct temporal properties. We provide full definitions and details in Appendix B.1.

### 3.4 Evaluation Metrics and Protocols

**Forecasting Accuracy.** We measure accuracy by comparing model predictions with ground truth before and after perturbation. We adopt three widely used scale-invariant metrics: *Normalized Mean Absolute Error (NMAE)*, *Normalized Root Mean Squared Error (NRMSE)*, and *Continuous Ranked Probability Score (CRPS)*. Formal definitions are provided in Appendix B.2.

**Adversarial Robustness.** Robustness is assessed by quantifying the change in forecasting performance under adversarial perturbations. This evaluation faces three challenges: (i) Ground truth is unavailable during attack construction, so we use the model's clean prediction as a surrogate target,

which introduces bias. (ii) Targeted and untargeted attacks have opposite objectives, necessitating a unified measure. (iii) time-series datasets can have very different scales, so robustness metrics must be insensitive to global rescaling to support fair cross-dataset comparison. To address these challenges, we propose the *Relative Error Deviation (RED$_\mathcal{E}$)*, which is a scale-invariant measure that captures the relative change in error in a direction-consistent manner. Let $\mathcal{E}(\cdot, \cdot)$ be a forecasting error metric (e.g., NMAE), and let $\hat{\mathbf{x}}^{\text{clean}}$, $\hat{\mathbf{x}}^{\text{adv}}$, and $\mathbf{y}$ denote the clean prediction, adversarial prediction, and reference sequence, respectively. We define

$$\text{RED}_\mathcal{E} = \frac{\mathcal{E}_{\text{attack}}}{\mathcal{E}^{\text{clean}} + \varepsilon}, \quad \mathcal{E}_{\text{attack}} = \begin{cases} \mathcal{E}^{\text{adv}} - \mathcal{E}^{\text{clean}}, & \text{untargeted attack}, \\ \mathcal{E}^{\text{clean}} - \mathcal{E}^{\text{adv}}, & \text{targeted attack}, \end{cases} \quad (7)$$

where $\mathcal{E}^{\text{clean}} = \mathcal{E}(\hat{\mathbf{x}}^{\text{clean}}, \mathbf{y})$, $\mathcal{E}^{\text{adv}} = \mathcal{E}(\hat{\mathbf{x}}^{\text{adv}}, \mathbf{y})$, and $\varepsilon > 0$ is a small constant. For untargeted attacks, it quantifies how strongly the forecast is *pushed away* from the clean prediction. For targeted attacks, it instead captures how closely the adversarial forecast *converges toward the attacker-specified target*, which can be particularly dangerous for downstream decision-making. Since RED$_\mathcal{E}$ can overstate attack impact when clean error is low, we complement it with *robustness curves* plotting absolute error against varying perturbation budgets. See details in Algorithm 1.

### 3.5 Defense Methods

As an initial step toward improving robustness, we explore two complementary defense strategies.

**Inference-Time Smoothing.** We employ a simple defense that suppresses high-frequency adversarial noise at test time using a moving-average filter. It operates directly on the input window without modifying $f_\phi$, requires no retraining, and introduces only a single hyperparameter (the kernel size). Given an input window $\mathbf{x}_{t-L:t}$ and a kernel size $W \in \mathbb{N}$, the smoothed series is defined as $\tilde{x}_{t-i} = \frac{1}{W_i} \sum_{m=0}^{W_i-1} x_{t-i-m}$, where $W_i = \min\{W, i+1\}$ ensures properly handles boundary cases.

**Adversarial Training (AT).** To enhance robustness more fundamentally, we fine-tune TSFMs with adversarial training in either latent space or input space. For latent adversarial training (LAT) (Casper et al., 2024), let $f_\phi = f_{\phi_2} \circ f_{\phi_1}$ denote the forecaster, where $f_{\phi_1}$ is a feature extractor and $f_{\phi_2}$ maps latents to outputs $\hat{\mathbf{y}}$. LAT introduces perturbations $\boldsymbol{\delta}^h$ in the latent representation and minimizes the worst-case forecasting loss under a bounded latent budget:

$$\min_\phi \ \mathbb{E}_{\mathbf{x} \sim \mathcal{P}(\mathcal{D}),\, t \sim \mathcal{P}(\mathcal{T})} \Big[ \max_{\|\boldsymbol{\delta}^h\|_p \leq \varepsilon} \sigma \cdot \mathcal{L}\big(f_{\phi_2}(f_{\phi_1}(\mathbf{x}_{t-L:t}) + \boldsymbol{\delta}^h), \mathbf{y}\big) \Big]. \quad (8)$$

In input-space adversarial training (IAT) (Madry et al., 2017), perturbations are instead applied directly to the input window. Given an input perturbation budget $\|\boldsymbol{\delta}^x\|_p \leq \varepsilon$, IAT optimizes

$$\min_\phi \ \mathbb{E}_{(\mathbf{x}, t)} \Big[ \max_{\|\boldsymbol{\delta}^x\|_p \leq \varepsilon} \mathcal{L}\big(f_\phi(\mathbf{x}_{t-L:t} + \boldsymbol{\delta}^x), \mathbf{y}\big) \Big], \quad (9)$$

where $\boldsymbol{\delta}^x$ is generated by projected gradient ascent and model parameters are updated on the resulting adversarial examples. Compared with LAT, IAT does not require access to intermediate representations but operates in the raw input space. Details for all defenses are provided in Appendix C.

## 4 Results and Analyses

In Section 4.1, we present overall robustness and attack results. Section 4.2 examines adversarial transferability and factors shaping attack impact. We further study robustness under different perturbation budgets and model sizes in Appendices E.7 and E.9. Section 4.3 evaluates two practical defenses for TSFMs, with additional analysis in Appendix E.11. Additional experimental settings and extended results are provided in Appendices D and E.

**Datasets.** We evaluate model robustness on eight datasets from the GIFT-Eval benchmark (Aksu et al., 2024), spanning diverse domains and sampling frequencies. Unless otherwise specified, all experiments are conducted in the short-term forecasting setting. Results under medium- and long-term prediction horizons are provided in Appendix E.10 for completeness. For short-term forecasting, we use a fixed input context length of 128 across all datasets. Additional dataset details, including domain characteristics and prediction lengths, are summarized in Table 6.

Table 2: **Untaregeted attacks against TSFMs.** For each model, $RED_{NMAE}$ is computed for every dataset and for all $4 \times 4$ budget combinations ($\epsilon \in \{0.25, 0.5, 0.75, 1.0\}$, $r \in \{0.25, 0.5, 0.75, 1.0\}$), then averaged over budgets and datasets. Red highlights the model with the largest average $RED_{NMAE}$ (most impacted). We provide detailed robustness curves across different perturbation levels in Appendix E.7. Clean forecasting performance is reported in Table 7.

| | **PGD** | | | | **SimBA** | | | | | |
| Dataset | TimesFM | TimeMoE | UniTS | Moirai | TimesFM | TimeMoE | UniTS | Moirai | Chronos | TabPFN-TS |
|---|---|---|---|---|---|---|---|---|---|---|
| Loop Seattle | 27.45 | 0.25 | 0.34 | 1.61 | 0.96 | 0.39 | 0.01 | 0.44 | 0.15 | 0.84 |
| BizITObs-L2C | 14.59 | 0.22 | 0.43 | 0.35 | 1.47 | 0.47 | 0.13 | 0.39 | 0.20 | 0.52 |
| Electricity | 27.25 | 0.19 | 0.18 | 0.31 | 1.45 | 0.37 | 0.00 | 0.06 | 0.05 | 0.53 |
| ETT1 | 32.45 | 0.20 | 0.58 | 1.63 | 1.66 | 0.80 | 0.06 | 0.63 | 0.50 | 1.36 |
| Hier. Sales | 45.28 | 0.08 | 1.46 | 1.16 | 3.99 | 0.71 | 0.23 | 0.40 | 0.58 | 2.17 |
| Jena Weather | 38.32 | 0.04 | 0.40 | 0.64 | 2.19 | 0.41 | 0.04 | 0.13 | 0.19 | 1.25 |
| Solar | 48.79 | 0.65 | 0.34 | 1.05 | 3.03 | 1.43 | 0.06 | 0.72 | 0.63 | 1.64 |
| US Births | 30.28 | 0.81 | 0.06 | 0.59 | 3.16 | 1.60 | -0.01 | 0.39 | 1.11 | 2.50 |

**Time-series Foundation Models.** We select six representative models for evaluation, including TimesFM (200M) (Das et al., 2024), TimeMoE (base) (Shi et al., 2025), UniTS (x128) (Gao et al., 2024), Moirai (base) (Woo et al., 2024), Chronos (small) (Ansari et al., 2024), and TabPFN-TS (Hoo et al., 2025) [2]. These models differ in architecture, decoding strategy, prediction head, and training paradigms. This diversity allows us to investigate how different design choices impact adversarial robustness. A detailed comparison of the models is provided in Appendix A.

**Evaluation Setup.** We adopt a deployment-matched setting: *zero-shot, frozen-checkpoint* inference. We report NMAE and $RED_{NMAE}$ as primary metrics. For black-box setting, we use SimBA (DCT basis) and ZOO (two-point finite differences) with step size $\alpha = 0.05$. For white-box, i.e. worst-case probe, we use PGD with the same $(\epsilon, r)$ budgets and $\alpha = 0.05$ for 300 iterations. All experiments are run on a single NVIDIA Tesla V100 GPU with CUDA 12.1. Metric definitions are provided in Appendix B.2, and defense implementation details in Appendix D.2.

### 4.1 Adversarial Robustness of TSFMs

**Subtle perturbations can cause severe degradation in forecast accuracy.** Our results show that most TSFMs are highly susceptible to adversarial attacks, with varying degrees of vulnerability across models. As shown in Table 2, TimesFM and TabPFN-TS exhibit particularly high sensitivity. Surprisingly, PGD attacks on TimesFM result in forecast errors up to 50 times higher than clean performance. For TabPFN-TS, the backbone was not originally trained for time series forecasting, potentially leading to poor generalization under perturbed conditions. Other models also exhibit significant degradation, highlighting adversarial vulnerability as a common concern in current TSFMs.

**Targeted behaviors can be induced even under small perturbation budgets.** As illustrated in Figure 1 and 3, targeted attacks can successfully manipulate TSFM outputs to match specific trajectories, even under small perturbation budgets. Interestingly, we find that attacks targeting the overall shape or trend of the forecast (e.g., scaling or shifting) are consistently more effective than those modifying only a local segment (e.g., local offset), as indicated by the higher RED scores in Tables 12 and 13. This suggests that TSFMs may lack strong global consistency constraints, making them easier to guide toward holistic patterns. In contrast, localized perturbations are often less effective, likely due to the model's reliance on contextual smoothing and local temporal dependencies. These internal dynamics make it difficult to isolate and alter a specific region without disrupting the broader sequence coherence. A robust model should resist both types of manipulations, particularly when perturbations are minimal, by preserving the structure of the predicted sequence.

**Disentangling Genuine Robustness from Gradient Obfuscation.** Among undefended models, TimeMoE and UniTS initially appear resilient to PGD attacks. For TimeMoE, our diagnostics suggest this resilience is illusory. While the MoE gating mechanism may disrupt gradient flow,

---
[2]Unless otherwise specified, we use the model size indicated in parentheses.

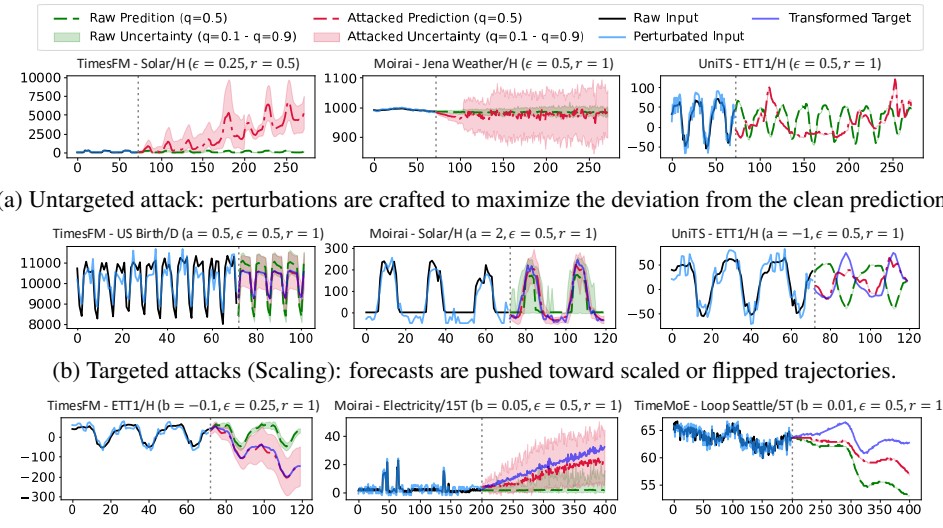

(a) Untargeted attack: perturbations are crafted to maximize the deviation from the clean prediction.

(b) Targeted attacks (Scaling): forecasts are pushed toward scaled or flipped trajectories.

(c) Targeted attacks (Drifting): perturbations induce gradual shifts in forecast trends.

Figure 3: **Visualization of untargeted and targeted adversarial attacks on TSFMs.** The $q$ indicates prediction quantile (e.g., $q = 0.5$ for median, $q = 0.1/0.9$ for uncertainty bands). The parameter $a$ controls the scaling of the target trajectory (with default $a = 1$), and $b$ controls the additive drift or offset (with default $b = 0$). Detailed $\text{RED}_{\text{NMAE}}$ scores are provided in Appendix E.8.

Table 3: **Robustness comparison between TSFMs and supervised models.** We report $\text{RED}_{\text{NMAE}}(\downarrow)$ under PGD untargeted attacks ($\epsilon = 0.5$, $r = 0.5$). More details see Appendix E.12.

| Dataset | TSFMs | | | Supervised | | |
|---|---|---|---|---|---|---|
| | TimesFM | Moirai | UniTS | GRU | TCN | Informer |
| ETTh1 | 2.1038 | 0.6814 | **0.0346** | 0.1456 | 0.0510 | 0.1302 |
| ETTh2 | 3.3649 | 0.5909 | 0.0587 | 0.1203 | **0.0043** | 0.0344 |
| ETTm1 | 6.5620 | 1.0180 | 0.1208 | 0.1087 | **0.0522** | 0.2252 |
| ETTm2 | 5.2638 | 1.0133 | 0.0798 | 0.1060 | **0.0023** | 0.1376 |
| Exchange | 4.2828 | 0.5000 | 0.0271 | 0.0537 | **0.0072** | 2.0443 |
| Weather | 5.2788 | 0.4640 | 0.1660 | 0.0422 | **0.0050** | 0.3578 |
| Electricity | 2.1000 | 0.6019 | 0.4733 | **0.0011** | 0.2314 | 0.0212 |

high vulnerability to one-step attacks (Appendix E.3) and black-box queries indicates that this is a case of *gradient obfuscation* rather than true robustness (Athalye et al., 2018). Conversely, UniTS demonstrates robustness across both white-box and black-box settings. We hypothesize this stems from its multi-task pretraining, which is known to enhance stability against distribution shifts and noise (Mao et al., 2020), though confirming this causal link requires further ablation. Overall, these case studies highlight that standard PGD evaluation can be misleading; architectural choices (like gating) and training strategies (like multi-tasking) significantly modulate adversarial behavior, necessitating rigorous diagnostic checks.

**Supervised models exhibit superior robustness compared to zero-shot TSFMs.** As detailed in Table 3, supervised models consistently outperform zero-shot TSFMs under identical perturbation budgets. This performance gap underscores a robustness–generalization trade-off: while TSFMs achieve universality through large-scale pretraining, they sacrifice the adversarial resilience inherent in models optimized for specific data distributions. Notably, the Transformer-based Informer is significantly more brittle than simpler architectures such as GRU and TCN. This aligns with established theoretical findings (Goodfellow et al., 2014; Ilyas et al., 2019) suggesting that high-dimensional, highly linearized models are inherently more susceptible to adversarial perturbations.

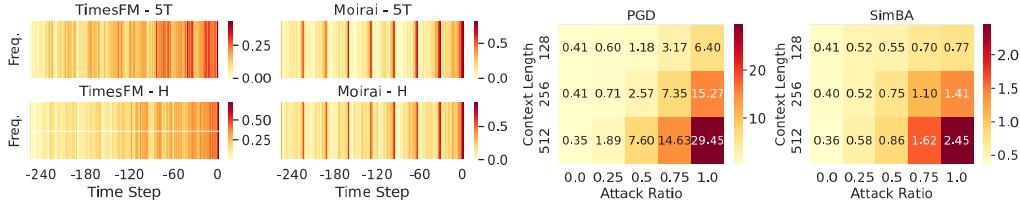

(a) Impact of perturbation locations. (NMAE).      (b) Impact of context length (NMAE).

Figure 4: **Effects of perturbation location and context length on adversarial robustness.** (a) Darker colors indicate higher frequencies at which a time point is identified as one of the most vulnerable. We compute the gradient magnitude wrt. the attack objective and select the top-25 positions as the most sensitive points. (b) NMAE under varying context lengths and attack ratios with PGD and SimBA attacks ($\epsilon = 0.5$). Higher values indicate greater performance degradation.

## 4.2 FURTHER ANALYSES

**Are perturbed series still the "same" series?** Time series are highly structure-sensitive (trend/seasonality), raising the question of whether adversarially perturbed inputs remain semantically consistent with the originals. We apply STL decomposition (Cleveland et al., 1990) to clean and perturbed inputs (PGD on TimesFM) and compute Pearson correlations between the corresponding trend, seasonal, and residual components. As shown in Table 8, these components remain highly correlated (typically $> 0.9$), indicating that global temporal structure is largely preserved. Nevertheless, such small, structured deviations are sufficient to induce substantial forecast degradation. This highlights a core insight of our work: TSFMs can fail even when perturbations preserve high-level structure, making detection difficult in the absence of semantic context.

**Failures are model-specific, with limited cross-model transfer.** Adversarial inputs crafted on one forecaster transfer poorly to others—even under untargeted objectives. From Table 11 we observe that PGD perturbations generated on TimesFM cause substantially smaller errors on Moirai and ARIMA than on the source model, indicating that the attack exploits model-specific inductive biases rather than generic distortions of the time series itself. However, limited transferability does not imply safety: query-based black-box attacks remain effective without source-model gradients, and architectural homogeneity can further increase transfer rates (Hwang et al., 2024).

**Points near the forecast boundary are most vulnerable.** We analyze input sensitivity by computing gradient magnitudes with respect to the attack objective and identifying the top-k most vulnerable input positions (Figure 4a). A clear pattern emerges: points closer to the forecast horizon consistently exhibit higher vulnerability. Additionally, vulnerability appears model-specific. For instance, Moirai shows consistent sensitivity at specific time positions across datasets, potentially due to its patch-based input configuration. These results suggest that defenses emphasizing boundary segments (e.g., recency-aware filtering or local regularization) may yield favorable robustness–utility trade-offs.

**Longer contexts enhance clean accuracy but amplify attack effects.** Table 4b shows a trade-off: longer input contexts boost clean accuracy but also enlarge the attack surface. Under PGD with $r = 0.5$, extending the context from 128 to 512 increases forecasting error by over 7×. Holding $r$ constant increases the absolute number of perturbed positions ($rL$), expanding the feasible set for the adversary; longer contexts also propagate perturbations over richer temporal dependencies. This is especially concerning for models like Moirai, which depend on long contexts, highlighting the need to balance accuracy and adversarial robustness.

## 4.3 EFFECTIVENESS OF DEFENSE STRATEGY

**Adversarial fine-tuning yields substantial worst-case robustness, even with out-of-domain data.** As detailed in Table 4, adversarial fine-tuning serves as a highly effective defense against strong white-box attacks. In-domain adversarial tuning (I-LAT) is particularly potent, reducing worst-case NMAE by $\sim$4–10× compared to vanilla models. Crucially, this robustness is highly transferable.

Table 4: **Defense results on TimesFM (NMAE↓)**. *Clean*: natural error (no attack). *no def.*: vanilla model. *(K=3/5)*: inference-time moving-average smoothing with kernel size $K$. *I-LAT*: latent adversarial training fine-tuned on the same dataset training split. *C-LAT*: cross-domain LAT. *C-IAT*: cross-domain input-space adversarial training. We use KDD Cup 2018 (Godahewa et al., 2021) for cross-domain training. See Appendix C for full results and more details.

| Dataset | Clean | PGD | | | | | | SimBA | | | | | |
|---|---|---|---|---|---|---|---|---|---|---|---|---|---|
| | | no def. | (K=3) | (K=5) | I-LAT | C-LAT | C-IAT | no def. | (K=3) | (K=5) | I-LAT | C-LAT | C-IAT |
| Loop Seattle | 0.113 | 1.213 | 0.416 | 0.399 | **0.132** | 0.154 | 0.170 | 0.167 | 0.154 | 0.121 | **0.109** | 0.128 | 0.141 |
| BizITObs-L2C | 2.904 | 19.947 | 14.325 | 8.331 | 4.397 | 5.085 | **4.092** | 6.495 | 3.853 | 3.897 | 3.773 | 3.860 | **3.281** |
| Electricity | 0.333 | 4.055 | 2.227 | 1.698 | 0.533 | **0.512** | 0.566 | 0.500 | **0.420** | 0.457 | 0.465 | 0.429 | 0.466 |
| ETT1 | 0.246 | 2.368 | 1.364 | 1.011 | **0.502** | 0.530 | 0.619 | 0.478 | **0.407** | 0.438 | 0.415 | 0.450 | 0.504 |
| Hier. Sales | 0.927 | 20.871 | 6.290 | 8.492 | **3.489** | 3.532 | 4.043 | **1.817** | 2.795 | 3.288 | 2.371 | 2.641 | 2.820 |
| Solar | 0.569 | 15.033 | 6.002 | 4.177 | **1.360** | 1.543 | 1.606 | 1.480 | 1.593 | 1.222 | **1.188** | 1.356 | 1.188 |
| US Birth | 0.033 | 0.237 | 0.205 | 0.301 | 0.120 | 0.132 | **0.110** | **0.072** | 0.105 | 0.111 | 0.097 | 0.113 | 0.090 |

Cross-domain variants (C-LAT), trained on completely unrelated datasets, retain 80–95While input-space adversarial training (IAT) follows similar trends, it is generally less stable than LAT. These findings highlight adversarial fine-tuning, particularly LAT, as a practical and low-cost approach for strengthening TSFMs in real deployments.

**Black-box robustness remains elusive.**  Unlike white-box scenarios, defending against query-based SimBA attacks proves significantly more challenging. Improvements from adversarial tuning are marginal and occasionally negative; for instance, LAT and IAT exacerbate vulnerability on datasets like Hierarchical Sales. This suggests a misalignment between the gradient-based optimization used in defenses and the query-based geometry of black-box attacks. Furthermore, inference-time smoothing is unreliable: while it mitigates PGD noise, it frequently degrades black-box robustness by indiscriminately suppressing high-frequency components that are critical for accurate forecasting.

**The trade-off between robustness and clean accuracy is dataset-dependent.**  As shown in Table 14, the impact of adversarial fine-tuning is non-uniform. On noisy datasets (e.g., Loop Seattle, BizITObs-L2C), adversarial perturbations appear to act as a regularizer, preserving or even improving clean performance. Conversely, on highly seasonal or low-variance datasets (e.g., Electricity, ETT1), both LAT and IAT compromise precision, introducing noticeable clean-error increases. Inference-time smoothing is similarly inconsistent: while occasionally beneficial, it more commonly degrades clean accuracy (Table 15) by filtering out legitimate high-frequency signal components.

## 5 DISCUSSION

This work establishes a rigorous, time-series–grounded framework for diagnosing adversarial risks in Time-Series Foundation Models (TSFMs). By focusing on native, non-LLM architectures, we fill a critical gap left by recent studies on LLM-based forecasters. Our analysis reveals that current TSFMs are alarmingly brittle: small, well-crafted perturbations can reliably steer forecasts toward malicious outcomes. Crucially, we uncover failure modes unique to this domain, such as horizon-proximal brittleness and a "context paradox". However, our findings also offer a promising path forward: we demonstrate that latent adversarial fine-tuning yields robust defenses that are highly transferable, remaining effective even when trained on out-of-domain data. This suggests that scalable, generalized safety measures are achievable for zero-shot deployment.

**Limitations and Future Directions.**  We prioritized widely used, deployment-feasible perturbations; future work should expand this scope to adaptive and universal strategies to fully map the threat landscape. Similarly, our defense analysis focused on lightweight methods, leaving resource-intensive techniques like adversarial pretraining as a direction to clarify robustness limits. Finally, the distinct vulnerability patterns we uncovered warrant deeper mechanistic analysis to isolate how architectural components, such as patching schemes, drive these failures.

## ETHICS STATEMENT

This work studies the adversarial robustness of time-series foundation models, and our experiments are conducted exclusively on publicly available benchmark datasets, without involving human subjects or private data. While our goal is to uncover and understand TSFM vulnerabilities, the techniques developed in this study could be misused. We therefore emphasize the importance of responsible use and advocate for parallel research into effective defense strategies.

## REPRODUCIBILITY STATEMENT

We take reproducibility seriously and provide all necessary resources to replicate our results. Our experiments are implemented in PyTorch, and we release the complete codebase, including attack-/defense implementations, evaluation protocols at `https://anonymous.4open.science/r/Attack_TSFMs-9622/`. Detailed descriptions of datasets, model configurations, and hyperparameter choices are included in the main text and Appendix D.

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

## USE OF LARGE LANGUAGE MODELS

We used large language models (LLMs) as assistive tools in two ways: (i) for improving the clarity and conciseness of manuscript writing, and (ii) for generating portions of experimental analysis code.

## A    DETAILS OF TIME SERIES FOUNDATION MODELS

Table 5: **Comparison of time-series foundation models used in this study.**

|  | **TimesFM** | **TimeMoE** | **UniTS** | **Moirai** | **Chronos** | **TabPFN** |
|---|---|---|---|---|---|---|
| Backbone | Dec-only | Dec-only | Enc-only | Enc-only | Enc-Dec | Enc-only |
| Decoding | AR | AR | NAR | NAR | AR | NAR |
| Pos. Emb. | Absolute PE | RoPE | Learnable PE | RoPE | Rel. PE | Calendar |
| Pred. Head | Point | Point | Point | Dist. | Dist. | Dist. |
| Loss | MSE | Huber+Aux | MSE | Likelihood | Cross-entropy | Cross-entropy |
| Patchify | ✓ | × | ✓ | ✓ | × | × |
| Standardization | ReVIN | ReVIN | ReVIN | ReVIN | Mean scaling | Z-score |
| Grad-based Att. | ✓ | ✓ | ✓ | ✓ | × | × |
| Pretrained Model | 200M
500M | 50M
200M | x32
x128 | small (13.8M)
base (91.4M)
large (311M) | tiny (8M)
mini (20M)
small (46M)
base (200M)
large (710M) | 11M |

**Backbone and Decoding Strategy.**    TimesFM (Das et al., 2024), Chronos (Ansari et al., 2024) and TimeMoE (Shi et al., 2025) adopt decoder-only Transformer architectures with autoregressive (AR) decoding, closely following the design of large language models. UniTS (Gao et al., 2024), TabPFN-TS (Hoo et al., 2025) and Moirai (Woo et al., 2024) use encoder-only backbones with non-autoregressive (NAR) decoding, enabling parallel prediction across time steps.

**Prediction Head and Loss.**    TimesFM, TimeMoE, and UniTS perform point forecasting optimized with MSE or Huber losses. In contrast, Moirai, Chronos, and TabPFN-TS adopt distributional forecasting heads trained with likelihood-based or cross-entropy losses.

**Input Patchification and Normalization.**    Most encoder-based models adopt input patchification to improve modeling efficiency, except Chronos and TabPFN. For standardization, all models except Chronos and TabPFN use ReVIN (Kim et al., 2022), a reversible instance normalization technique tailored for time series. Chronos applies mean scaling, while TabPFN uses z-score normalization.

**Gradient-based Attack Compatibility.**    Not all models are compatible with gradient-based white-box attacks. TimesFM, TimeMoE, UniTS, and Moirai produce continuous outputs, allowing direct optimization through gradient-based methods such as PGD. In contrast, Chronos and TabPFN-TS discretize both inputs and outputs by treating time points as tokens and generating categorical distributions over predefined bins. This discretization renders standard gradient-based attacks impractical.

## B    DETAILS OF EVALUATION PROTOCOLS

Algorithm 1 summarizes our evaluation procedure, where the attacker perturbs the input sequence $\mathbf{x}_{t-L:t}$ within a constrained budget to either increase forecast error (untargeted) or drive the output toward a predefined target trajectory (targeted).

### B.1    ATTACKING STRATEGIES

**Fast Gradient Sign Method (FGSM)**    FGSM (Goodfellow et al., 2015) is a single-step white-box attack that perturbs the input once in the direction of the gradient of the loss with respect to the input.

**Algorithm 1** TSFM Adversarial Robustness Evaluation
________________________________________________

**Require:** Pre-trained model $f_\phi$, input $\mathbf{x}_{t-L:t}$, budget $(r, \epsilon)$, attack type $\sigma \in \{+1, -1\}$, transform
     params $(a, c(\cdot))$
1: $\hat{\mathbf{x}}^{\text{clean}} \leftarrow f_\phi(\mathbf{x}_{t-L:t})$                                                   ▷ Sec. 3.1
2: **if** $\sigma = +1$ **then**                                                         ▷ untargeted
3:      $\mathbf{y} \leftarrow \hat{\mathbf{x}}^{\text{clean}}$
4: **else**                                                                   ▷ targeted
5:      $\mathbf{y} \leftarrow \mathcal{M}(\hat{\mathbf{x}}^{\text{clean}}; a, c)$                                    ▷ Eq. equation 3
6: **end if**
7: $\mathcal{S} \leftarrow \{\boldsymbol{\delta} : \|\boldsymbol{\delta}\|_0 \leq rL, \|\boldsymbol{\delta}\|_\infty \leq \epsilon\}$                     ▷ Eq. equation 4
8: $\boldsymbol{\delta}^* \leftarrow \text{ATTACK}(f_\phi, \mathbf{x}_{t-L:t}, \mathbf{y}, \mathcal{S}, \sigma)$                   ▷ Sec. 3.3
9: $\hat{\mathbf{x}}^{\text{adv}} \leftarrow f_\phi(\mathbf{x}_{t-L:t} + \boldsymbol{\delta}^*)$
10: $\text{RED}_\mathcal{E} \leftarrow \text{COMPUTERED}(\hat{\mathbf{x}}^{\text{clean}}, \hat{\mathbf{x}}^{\text{adv}}, \mathbf{y})$            ▷ Eq. equation 7
11: **return** $\hat{\mathbf{x}}^{\text{adv}}, \text{RED}_\mathcal{E}$
________________________________________________

The update rule is given by:

$$\mathbf{x}^{\text{adv}} = \mathbf{x} + \epsilon \cdot \text{sign}\left(\nabla_\mathbf{x} \mathcal{L}(f_\phi(\mathbf{x}), y)\right), \tag{10}$$

where $\mathcal{L}$ is the loss function, $f_\phi$ is the model, and $\epsilon$ is the perturbation budget controlling the maximum allowed perturbation.

**Projected Gradient Descent (PGD)**    PGD (Madry et al., 2017) is a white-box attack that iteratively perturbs the input in the direction of the gradient of the loss with respect to the input. At each step, the perturbation is projected back onto the feasible $\ell_p$-norm ball to ensure it stays within the allowed budget. The update rule is given by:

$$\mathbf{x}^{k+1} = \Pi_{\mathcal{B}_p(\mathbf{x}, \epsilon)}\left(\mathbf{x}^k + \alpha \cdot \text{sign}\left(\nabla_\mathbf{x} \mathcal{L}(f_\phi(\mathbf{x}^k), y)\right)\right), \tag{11}$$

where $\Pi$ denotes the projection operator, $\mathcal{L}$ is the loss function, $f_\phi$ is the model, $\epsilon$ is the perturbation budget, and $\alpha$ is the step size.

**Zeroth Order Optimization (ZOO)**    ZOO (Chen et al., 2017) is a black-box attack that approximates the gradient using only function evaluations, without requiring access to the model's internals. The directional derivative is estimated using finite differences:

$$\frac{\partial \mathcal{L}}{\partial x_i} \approx \frac{\mathcal{L}(\mathbf{x} + h\mathbf{e}_i) - \mathcal{L}(\mathbf{x} - h\mathbf{e}_i)}{2h}, \tag{12}$$

where $h$ is a small constant, and $\mathbf{e}_i$ is the standard basis vector in the $i$-th direction. The estimated gradients are then used to perform gradient-based optimization in the black-box setting.

**Simple Black-box Attack (SimBA).**    SimBA (Guo et al., 2019) is a decision-based black-box attack that perturbs the input along randomly selected directions from an orthonormal basis. At each iteration, a candidate direction $\mathbf{q} \in Q$ and step size $\alpha > 0$ are selected based on attack objective $g_\phi$:

$$\boldsymbol{\delta}^{k+1} = \begin{cases} \boldsymbol{\delta}^k + \alpha\mathbf{q}, & \text{if } g_\phi(\boldsymbol{\delta}^k + \alpha\mathbf{q}) > g_\phi(\boldsymbol{\delta}^k), \\ \boldsymbol{\delta}^k - \alpha\mathbf{q}, & \text{if } g_\phi(\boldsymbol{\delta}^k - \alpha\mathbf{q}) > g_\phi(\boldsymbol{\delta}^k), \\ \boldsymbol{\delta}^k, & \text{otherwise.} \end{cases} \tag{13}$$

To ensure efficiency, SimBA samples directions without replacement and guarantees a bounded $\ell_2$-norm of the perturbation: $\|\delta\|_2 = \sqrt{T}\alpha$ after $T$ updates. Its only hyperparameters are the set of orthonormal vectors $Q$ and the step size $\epsilon$.

1. Cartesian Basis (Point-wise): The *standard basis* $Q_{\text{ID}} = \{\mathbf{e}_1, \ldots, \mathbf{e}_L\}$ consists of unit vectors, where each $\mathbf{e}_i \in \mathbb{R}^L$ has a 1 at the $i$-th position and zeros elsewhere. This basis corresponds to perturbing individual time points independently. Each attack iteration modifies a single, randomly selected time step by increasing or decreasing its value.

2. Discrete Cosine Transform (DCT) Basis: The *DCT basis* $Q_{\text{DCT}}$ is an orthonormal set of vectors that transform a time-domain signal $x \in \mathbb{R}^L$ into a sequence of frequency coefficients. To encourage smooth and perceptually coherent perturbations, we restrict the perturbation to lie within a low-frequency subspace. Specifically, we retain only a fraction $r \in (0, 1]$ of the lowest-frequency components from $Q_{\text{DCT}}$.

3. Wavelet Basis: The *wavelet basis* $Q_{\text{WAV}}$ is derived from a discrete wavelet transform (DWT), such as the Haar family. This basis provides a time-frequency decomposition, where each vector captures information localized in both time and scale (resolution). Perturbations in this basis can target specific trends or local details. For control over the granularity of perturbation, we optionally restrict $Q_{\text{WAV}}$ to low-frequency (approximation) coefficients at a specified decomposition level $\ell$.

## B.2 METRICS FOR ACCURACY EVALUATION

**The Normalized Mean Absolute Error (NMAE)**    The NMAE (Yu et al., 2016) is a normalized version of the MAE, which is dimensionless and facilitates the comparability of the error magnitude across different datasets or scales. The mathematical representation of NMAE is given by:

$$\text{NMAE} = \frac{\sum_{t=1}^{T} |x_t - \hat{x}_t|}{\sum_{t=1}^{T} |x_t|}. \tag{14}$$

**Normalized Root Mean Squared Error (NRMSE)**    The NRMSE is a normalized version of the Root Mean Squared Error (RMSE), which quantifies the average squared magnitude of the error between forecasts and observations, normalized by the expectation of the observed values. It can be formally written as:

$$\text{NRMSE} = \frac{\sqrt{\frac{1}{T} \sum_{t=1}^{T} (x_t - \hat{x}_t)^2}}{\frac{1}{T} \sum_{t=1}^{T} |x_t|}. \tag{15}$$

**Continuous Ranked Probability Score (CRPS)**    The CRPS (Matheson & Winkler, 1976) quantifies the agreement between a cumulative distribution function (CDF) $F$ and an observation $x$, represented as:

$$\text{CRPS} = \int_{\mathbb{R}} (F(z) - \mathbb{I}\{x \le z\})^2 dz, \tag{16}$$

where $\mathbb{I}\{x \le z\}$ denotes the indicator function, equating to one if $x \le z$ and zero otherwise.

Being a proper scoring function, CRPS reaches its minimum when the predictive distribution $F$ coincides with the data distribution. When using the empirical CDF of $F$, denoted as $\hat{F}(z) = \frac{1}{n} \sum_{i=1}^{n} \mathbb{I}\{X_i \le z\}$, where $n$ represents the number of samples $X_i \sim F$, CRPS can be precisely calculated from the simulated samples of the conditional distribution $p_\theta(\boldsymbol{x}_t | \boldsymbol{h}_t)$. In our practice, 100 samples are employed to estimate the empirical CDF.

## C  DETAILS OF DEFENSE METHODS

### C.1  INFERENCE-TIME SMOOTHING

Inference-Time Smoothing is a purely test-time defense that attenuates high-frequency by applying a simple temporal filter to the input window. It does not modify $f_\phi$, requires no retraining, and introduces no variants beyond a single filter with one hyperparameter. Given an input window $\mathbf{x}_{t-L:t} = (x_{t-L+1}, \ldots, x_t)$ and a window size $W \in \mathbb{N}$, we define the smoothed window

$$\tilde{x}_{t-i} = \frac{1}{W_i} \sum_{m=0}^{W_i - 1} x_{t-i-m}, \qquad i = 0, \ldots, L - 1, \tag{17}$$

where $W_i = \min\{W, i + 1\}$ ensures causality and handles left-boundary samples (i.e., partial averages when fewer than $W$ past values are available). The smoothed forecast is obtained by a single forward pass:

$$\hat{\mathbf{x}}_{t+1:t+T} = f_\phi(\tilde{\mathbf{x}}_{t-L:t}). \tag{18}$$

This operation is a causal low-pass filter that suppresses small, rapid oscillations typical of adversarial noise while preserving local trend/seasonality.

## C.2 LATENT ADVERSARIAL TRAINING

To better address unforeseen failure modes, we also explore fine-tune the TSFMs using the Latent Adversarial Training (LAT). Unlike standard adversarial training that perturbs inputs, LAT applies perturbations in the model's latent space (Casper et al., 2024). Specifically, let the forecaster decompose as $f_\phi = f_{\phi_2} \circ f_{\phi_1}$ with latent $\boldsymbol{h} = f_{\phi_1}(\mathbf{x}_{t-L:t})$ and prediction $\hat{\mathbf{y}} = f_{\phi_2}(\boldsymbol{h})$. Given a loss $\mathcal{L}$ and an $\ell_p$-budget $\|\boldsymbol{\delta}^h\|_p \leq \varepsilon$, LAT trains the model to minimize the worst-case forecasting loss under bounded *latent* perturbations:

$$\min_{\phi} \; \mathbb{E}_{(\mathbf{x},t)}\Big[ \max_{\|\boldsymbol{\delta}^h\|_p \leq \varepsilon} \; \mathcal{L}\big(f_{\phi_2}(\boldsymbol{h} + \boldsymbol{\delta}^h), \mathbf{y}\big)\Big]. \tag{19}$$

In practice, we solve the inner maximization by projected gradient *ascent* on $\boldsymbol{\delta}^h$ and the outer minimization by gradient *descent* on $\phi$ (Alg. 2). To avoid drifting into irrelevant activation ranges, the perturbed activations are clipped to the batchwise range of the unperturbed latent. The formulation also accommodates targeted or untargeted objectives via a direction parameter $\sigma \in \{+1, -1\}$ in the inner loss.

---

**Algorithm 2** Latent Adversarial Training (LAT) for Forecasting

---

**Require:** Univariate series $\mathbf{x}_{1:T}$; window length $L$, horizon $T$. Model parameter $\phi = (\phi_1, \phi_1)$, feature extractor $f_{\phi_1}$, latent-to-output mapping $f_{\phi_2}$; loss $\mathcal{L}$; attack direction $\sigma \in \{+1, -1\}$; budget $(r, \varepsilon)$; inner steps $T_\delta$; inner/outer rates $(\eta_\delta, \eta_\phi)$.
1: **for** each minibatch $\mathcal{B} = \{(\mathbf{x}_{t-L:t}, \mathbf{y})\}$ **do**
2:     Compute the latent representation: $\boldsymbol{h} \leftarrow f_{\phi_1}(\mathbf{x}_{t-L:t})$
3:     Initialize $\boldsymbol{\delta}^h \sim \mathcal{N}(0, I)$; $\boldsymbol{\delta}^h \leftarrow \mathrm{Proj}_{\mathcal{S}_h}(\boldsymbol{\delta}^h)$
4:     **for** $\tau = 1, \ldots, T_\delta$ **do**                                   ▷ inner ascent
5:         $\hat{\mathbf{y}}^{\mathrm{adv}} \leftarrow f_{\phi_2}(\boldsymbol{h} + \boldsymbol{\delta}^h)$
6:         Compute the adversarial objective: $\mathcal{L}_{\mathrm{adv}} \leftarrow \frac{1}{|\mathcal{B}|} \sum \sigma \cdot \mathcal{L}(\hat{\mathbf{y}}^{\mathrm{adv}}, \mathbf{y})$
7:         Update the perturbation via gradient ascent: $\boldsymbol{\delta}^h \leftarrow \boldsymbol{\delta}^h + \eta_\delta \nabla_{\boldsymbol{\delta}^h} \mathcal{L}_{\mathrm{adv}}$
8:         $\boldsymbol{\delta}^h \leftarrow \mathrm{Proj}_{\mathcal{S}_h}(\boldsymbol{\delta}^h)$
9:         $\boldsymbol{h} + \boldsymbol{\delta}^h \leftarrow \mathrm{clip}(\boldsymbol{h} + \boldsymbol{\delta}^h, \; \min(\boldsymbol{h}), \max(\boldsymbol{h}))$
10:    **end for**
11:    $\hat{\mathbf{y}}^{\mathrm{adv}} \leftarrow f_{\phi_2}(\boldsymbol{h} + \boldsymbol{\delta}^h)$
12:    Loss with adversarial perturbation: $\mathcal{L}_{\mathrm{total}} \leftarrow \frac{1}{|\mathcal{B}|} \sum \mathcal{L}(\hat{\mathbf{y}}^{\mathrm{adv}}, \mathbf{y})$
13:    $\phi \leftarrow \phi - \eta_\phi \nabla_\phi \mathcal{L}_{\mathrm{total}}$                              ▷ outer descent
14: **end for**

---

## C.3 INPUT-SPACE ADVERSARIAL TRAINING

In addition to smoothing and latent-space adversarial training, we also consider a conventional input-space adversarial training (IAT) baseline. IAT follows the classic formulation of adversarial training (Madry et al., 2017), where the model is optimized to minimize the forecasting loss under worst-case perturbations applied directly to the input window $\mathbf{x}_{t-L:t}$.

Given a perturbation budget $\|\boldsymbol{\delta}^x\|_p \leq \varepsilon$ and a loss $\mathcal{L}$, IAT solves the min–max problem

$$\min_{\phi} \; \mathbb{E}_{(\mathbf{x},t)}\Big[ \max_{\|\boldsymbol{\delta}^x\|_p \leq \varepsilon} \; \mathcal{L}\big(f_\phi(\mathbf{x}_{t-L:t} + \boldsymbol{\delta}^x), \mathbf{y}\big)\Big]. \tag{20}$$

We use projected gradient ascent to generate input perturbations and gradient descent to update model parameters. At each iteration, the adversarial example is formed as $\mathbf{x}^{\mathrm{adv}} = \mathbf{x} + \boldsymbol{\delta}^x$, and parameter updates follow:

$$\phi \; \leftarrow \; \phi - \eta_\phi \nabla_\phi \mathcal{L}\big(f_\phi(\mathbf{x}^{\mathrm{adv}}), \mathbf{y}\big). \tag{21}$$

In our experiments, we use the same learning schedules and fine-tuning settings as in LAT to ensure a fair comparison.

## D ADDITIONAL DETAILS OF EXPERIMENT SETTING

### D.1 DATASET DETAILS

We adopt benchmark datasets from the GIFT-Eval benchmark (Aksu et al., 2024), which includes a diverse set of real-world time-series datasets covering multiple domains, sampling granularities, and forecasting settings. For this study, we select a subset of these datasets to ensure broad domain coverage. A complete summary of the dataset characteristics, including domain, number of target variables, number of series, sampling frequency, input windows, and prediction lengths, is provided in Table 6.

Table 6: **Summary of datasets used in our experiments.** Datasets such as Solar, Electricity, and ETT support short-, medium-, and long-term forecasting settings, while others like US Births and Hierarchical Sales are limited to short-term prediction scenarios.

| Dataset | Domain | #Target Var | # Series | Frequency | # Windows | Pred Len |
|---------|--------|-------------|----------|-----------|-----------|----------|
| Solar | Energy | 1 | 137 | 10T
H | 20/11/8
29/2/2 | 48/480/720
48/480/720 |
| Electricity | Energy | 1 | 370 | 15T
H | 20/20/20
20/8/5 | 48/480/720
48/480/720 |
| ETT1 | Energy | 7 | 1 | 15T
H | 20/15/10
20/4/3 | 48/480/720
48/480/720 |
| Loop Seattle | Transport | 1 | 323 | 5T
H | 20/20/15
19/2/2 | 48/480/720
48/480/720 |
| BizTObs - L2C | Web/CloudOps | 7 | 1 | 5T
H | 20/7/5
6/1/1 | 48/480/720
48/480/720 |
| Jena Weather | Nature | 21 | 1 | 10T
H | 20/11/8
19/2/2 | 48/480/720
48/480/720 |
| US Births | Healthcare | 1 | 1 | D/W/M | 20/14/2 | 30/8/12 |
| Hierarchical Sales | Sales | 1 | 118 | D/W | 7/4 | 30/8 |

### D.2 IMPLEMENTATION DETAILS OF DEFENSE

For inference-time smoothing, we apply a moving-average filter with kernel sizes $K \in \{3, 5, 7\}$. This is a pre-processing step applied at inference, requires no retraining, and introduces only a single hyperparameter ($K$). For latent adversarial training, we fine-tune the model for 5 epochs using Adam with a learning rate of $1 \times 10^{-4}$. The latent perturbation budget is set to $\epsilon = 0.5$ with $\ell_\infty$ constraints, and adversarial perturbations are optimized for 5 inner steps per batch. Fine-tuning is performed on the training split of each dataset unless otherwise noted (cross-domain experiments use KDD Cup 2018). We fine-tune with batch size 64.

## E ADDITIONAL EXPERIMENTAL RESULTS

### E.1 PERFORMANCE COMPARISON OF TSFMS

In Table 7, we present the forecasting performance of six TSFMs on unperturbed inputs across eight datasets. Among all models, TimesFM consistently achieves strong zero-shot forecasting performance across diverse domains, followed closely by Moirai. This highlights the benefit of large-scale pretraining and architectural generality. However, our robustness evaluation reveals that stronger predictive accuracy does not necessarily imply higher adversarial resilience. In fact, these high-performing models often exhibit greater vulnerability to adversarial perturbations. This observation underscores a critical challenge: how to effectively balance predictive accuracy and robustness in the design of TSFMs. Addressing this trade-off remains an open and urgent research direction.

Table 7: **Raw forecasting performance of TSFMs across multiple datasets.** All results are reported under the short-term setting with context length 128. We evaluate each model using three metrics: NMAE, NRMSE, and CRPS. The best result for each metric is **bolded**, and the second best is underlined.

| Dataset | Chronos | | | Moirai | | | TabPFN-TS | | | TimeMoE | | | TimesFM | | | UniTS | | |
|---|---|---|---|---|---|---|---|---|---|---|---|---|---|---|---|---|---|---|
| | NMAE | NRMSE | CRPS | NMAE | NRMSE | CRPS | NMAE | NRMSE | CRPS | NMAE | NRMSE | CRPS | NMAE | NRMSE | CRPS | NMAE | NRMSE | CRPS |
| Loop Seattle | 0.08 | 0.10 | 0.13 | 0.07 | **0.07** | **0.10** | 0.10 | 0.10 | 0.13 | **0.06** | 0.08 | 0.11 | 0.07 | 0.08 | 0.11 | 0.07 | 0.08 | 0.12 |
| BizITObs-L2C | 1.16 | 1.42 | 1.86 | 1.19 | 1.19 | 1.54 | 1.43 | 1.43 | 1.77 | 1.22 | 1.46 | 1.99 | **0.91** | **1.00** | 1.55 | 1.04 | 1.29 | 1.80 |
| Electricity | 0.28 | 0.33 | 0.45 | 0.27 | **0.27** | **0.38** | 0.34 | 0.34 | 0.45 | 0.28 | 0.34 | 0.45 | 0.27 | 0.32 | 0.45 | 0.28 | 0.34 | 0.46 |
| ETT1 | 0.23 | 0.27 | 0.52 | 0.27 | 0.27 | 0.48 | 0.33 | 0.33 | 0.56 | 0.22 | 0.27 | 0.50 | 0.23 | 0.27 | 0.52 | **0.20** | **0.25** | **0.46** |
| Hierarchical Sales | **0.75** | **0.89** | **1.66** | 1.29 | 1.29 | 1.89 | 1.44 | 1.44 | 2.13 | 0.87 | 1.00 | 1.78 | 0.80 | 0.95 | 1.79 | 0.83 | 0.98 | 1.80 |
| Jena Weather | 0.05 | 0.06 | 0.22 | 0.21 | 0.21 | 0.34 | 0.08 | 0.08 | 0.26 | 0.06 | 0.07 | 0.28 | 0.05 | 0.06 | 0.23 | **0.05** | **0.06** | **0.21** |
| Solar | 0.50 | 0.59 | 0.96 | 0.41 | 0.41 | 0.71 | 0.92 | 0.92 | 1.19 | 0.41 | 0.52 | 0.97 | **0.36** | **0.40** | 0.76 | 0.42 | 0.49 | 0.87 |
| US Births | 0.03 | 0.03 | 0.04 | 0.03 | 0.03 | 0.04 | 0.10 | 0.10 | 0.12 | 0.04 | 0.05 | 0.06 | **0.02** | **0.03** | **0.04** | 0.02 | 0.03 | 0.04 |

Table 8: **Structural similarity between clean and adversarial inputs.** Attacks use $\epsilon = 0.25, r = 0.5$. We report Pearson correlations of seasonal, trend, and residual components after STL decomposition, along with NMAE on clean vs. attacked series. High correlations ($> 0.9$ in most cases) indicate that global structure is preserved.

| Dataset | Season Corr. | Trend Corr. | Resi. Corr. | NMAE (Raw / Attacked) |
|---|---|---|---|---|
| US Birth/D | 0.9727 | 0.9686 | 0.9856 | 0.0345 / 0.1226 |
| Loop Seattle/H | 0.9786 | 0.9424 | 0.9195 | 0.0732 / 0.1407 |
| Electricity/H | 0.9131 | 0.8938 | 0.8968 | 0.2703 / 0.6824 |
| ETT1/H | 0.9503 | 0.9659 | 0.9007 | 0.0604 / 0.2908 |

## E.2 STRUCTURAL CONSISTENCY UNDER ADVERSARIAL PERTURBATIONS

**Adversarial perturbations preserve structure yet degrade forecasts.** Table 8 quantifies the effect of adversarial perturbations on temporal structure. Across datasets, the seasonal, trend, and residual components of clean and attacked inputs remain highly correlated, typically above 0.9. This indicates that adversarial perturbations do not fundamentally distort the global signal structure, which would make them difficult to detect with simple statistical checks. Overall, these results highlight a critical challenge: TSFMs can fail catastrophically under perturbations that preserve high-level structure, making adversarial inputs both effective and stealthy.

## E.3 SINGLE-STEP ATTACK: FGSM RESULTS

To further examine whether the apparent robustness of sparse MoE-style architectures arises from gradient obfuscation (Athalye et al., 2018), we evaluate the single-step FGSM. Table 9 reports $\text{RED}_{\text{NMAE}}$ under an untargeted FGSM with $\epsilon = 0.5$ and $r = 0.5$.

On average across six datasets, TimeMoE does not consistently maintain robustness: it achieves the lowest error on only 2/6 datasets (BizITObs-L2C and Hier. Sales), while TimesFM is strongest on 4/6 (Loop Seattle, Electricity, ETT1, US Births). These single-step results contrast with the PGD-based trend in the main text and align with the gradient-obfuscation interpretation: when gradients are partially disrupted by MoE gating, multi-step methods like PGD can be deceptively weaker, while single-step (and black-box) attacks reveal more severe vulnerabilities.

## E.4 ADDITIONAL METRICS FOR EVALUATION

As a supplement to Table 2, we report the $\text{RED}_{\text{CRPS}}$ scores under untargeted attacks in Table 10. CRPS is a widely used metric for evaluating the quality of probabilistic forecasts. We observe that the robustness rankings across models based on CRPS are largely consistent with those based on NMAE.

## E.5 EFFECTIVENESS OF ATTACK STRATEGIES

**All TSFMs are vulnerable to adversarial perturbations, with varying degrees of susceptibility.**
Figure 5a compares the effectiveness of different attack strategies across various TSFMs under a fixed perturbation budget (untargeted attacks, $\epsilon = 0.5$, $r = 1$). Among the attack methods, PGD consistently achieves the strongest performance, followed by SimBA and then ZOO. For SimBA

Table 9: **Adversarial vulnerability under a single-step FGSM attack.** We report $\text{RED}_{\text{NMAE}}$ ($\downarrow$) for untargeted FGSM with $\epsilon = 0.5$ and $r = 0.5$. Lower values indicate stronger robustness.

| Dataset | TimesFM | Moirai | TimeMoE |
|---|---|---|---|
| Loop Seattle | 0.001770 | 0.186896 | 0.094545 |
| BizITObs-L2C | 0.055058 | 0.044279 | 0.016831 |
| Electricity | -0.013217 | 0.012086 | 0.087647 |
| ETT1 | 0.098068 | 0.319565 | 0.107615 |
| Hier. Sales | 0.153016 | 0.776197 | 0.049535 |
| US Births | 0.030395 | 0.077895 | 0.212389 |

Table 10: **Untaregeted attacks against TSFMs.** We report the $\text{RED}_{\text{CRPS}}$ averaged across attack budgets ($\epsilon \in \{0.25, 0.5, 0.75, 1\}$, $r \in \{0.25, 0.5, 0.75, 1\}$) and datasets. Red is used to denote the model most impacted by the attack.

| Dataset | PGD | | | | SimBA (Wavelet) | | | | | |
|---|---|---|---|---|---|---|---|---|---|---|
| | TimesFM | TimeMoE | UniTS | Moirai | TimesFM | TimeMoE | UniTS | Moirai | Chronos | TabPFN-TS |
| Loop Seattle | 27.35 | 0.25 | 0.34 | 1.74 | 0.93 | 0.39 | 0.01 | 0.41 | 0.13 | 0.82 |
| BizITObs-L2C | 15.39 | 0.22 | 0.43 | 0.41 | 1.65 | 0.47 | 0.13 | 0.35 | 0.15 | 0.53 |
| Electricity | 27.45 | 0.19 | 0.18 | 0.35 | 1.42 | 0.37 | 0.00 | 0.05 | 0.03 | 0.48 |
| ETT1 | 32.80 | 0.20 | 0.58 | 1.69 | 1.68 | 0.80 | 0.06 | 0.59 | 0.54 | 1.37 |
| Hierarchical Sales | 44.72 | 0.08 | 1.46 | 1.04 | 3.87 | 0.71 | 0.23 | 0.35 | 0.51 | 2.09 |
| Jena Weather | 37.87 | 0.04 | 0.40 | 0.63 | 2.12 | 0.41 | 0.04 | 0.10 | 0.24 | 1.11 |
| Solar | 48.11 | 0.65 | 0.34 | 1.09 | 3.03 | 1.43 | 0.06 | 0.78 | 0.59 | 1.47 |
| US Births | 30.59 | 0.81 | 0.06 | 0.70 | 3.35 | 1.60 | -0.01 | 0.41 | 1.06 | 2.47 |

variants, the choice of perturbation basis significantly impacts effectiveness: wavelet-based directions perform best, followed by point-wise and DCT bases. These results highlight both the vulnerability of current TSFMs and the importance of attack design choices, including basis structure and optimization method, in determining attack success.

**Gradient-based attacks are strong, but not always sufficient.** PGD generally outperforms black-box methods, while SimBA is moderately effective and ZOO is weakest (Appendix E.5). However, their effectiveness is not universal. Models like Chronos and TabPFN-TS apply input discretization, limiting gradient accessibility, while TimeMoE's gating mechanism in its mixture-of-experts architecture may disrupt gradient flow, weakening the impact of gradient-based attacks. In addition, the choice of perturbation basis influences attack strength. Figure 5b shows that the wavelet-based perturbations outperform DCT and point-wise strategies. These results highlight the need to align attack strategies with model architecture and data properties.

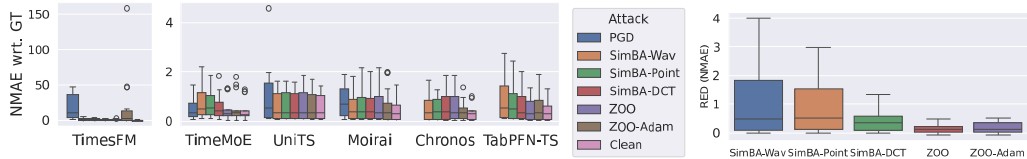

(a) Attack performance across TSFMs. TimesFM is shown separately due (b) Comparison of black-box attacks. to its large performance variance under PGD.

Figure 5: **Effectiveness of untargeted adversarial attacks across different strategies.** We evaluate PGD, SimBA (with wavelet, point, and DCT bases), and ZOO (with standard and Adam optimizers) under a fixed budget of $\epsilon = 0.5$, $r = 1$. Results are averaged over all datasets.

### E.6 CROSS-MODEL TRANSFERABILITY OF ADVERSARIAL EXAMPLES

The results in Table 11 show that adversarial perturbations crafted on TimesFM transfer very poorly to other forecasters, highlighting that adversarial vulnerabilities are highly model-specific rather than

Table 11: **Cross-model transferability of PGD adversarial examples.** Adversarial inputs are generated on TimesFM and evaluated on other models. We report NMAE for clean and perturbed inputs, and RED$_{\text{NMAE}}$ ($\downarrow$); lower values indicate weaker transferability.

| Dataset | TimesFM | | | $\rightarrow$ Moirai | | | $\rightarrow$ ARIMA | | |
|---|---|---|---|---|---|---|---|---|---|
| | Clean | Perturb | RED | Clean | Perturb | RED | Clean | Perturb | RED |
| Loop Seattle | 0.073 | 0.310 | 3.236 | 0.095 | 0.110 | 0.154 | 0.095 | 0.098 | 0.039 |
| BizITObs-L2C | 1.015 | 9.401 | 6.564 | 1.591 | 1.779 | 0.118 | 1.402 | 1.360 | -0.030 |
| Electricity | 0.328 | 5.707 | 16.373 | 0.410 | 0.459 | 0.121 | 0.337 | 0.341 | 0.011 |
| ETT1 | 0.270 | 4.044 | 13.961 | 0.284 | 0.403 | 0.419 | 0.446 | 0.455 | 0.021 |
| Hier. Sales | 0.912 | 2.568 | 1.816 | 1.577 | 2.420 | 0.535 | 1.521 | 1.594 | 0.048 |
| US Births | 0.034 | 0.123 | 2.558 | 0.051 | 0.059 | 0.154 | 0.103 | 0.104 | 0.005 |

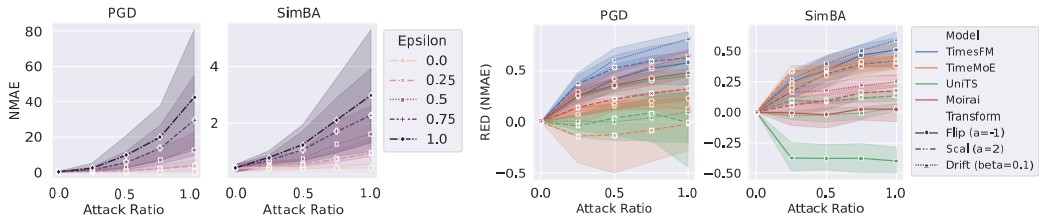

(a) Robustness curve in untargeted setting.  (b) Robustness curve in targeted setting.

Figure 6: **Impact of perturbation budgets.** (a) For untargeted attacks, we report NMAE across varying attack budgets (i.e., $r$ and $\epsilon$). (b) For targeted attacks, we use RED$_{\text{NMAE}}$ to measure alignment between the perturbed prediction and the target, where higher values indicate more successful attacks.

generic input distortions. RED$_{\text{NMAE}}$ values drop sharply when PGD examples from TimesFM are evaluated on Moirai and ARIMA. This indicates that adversarial perturbations exploit model-specific vulnerabilities tied to architectural and training biases. Adversarial examples do not generalize across models, which may limit the threat of universal adversarial attacks but also suggests that robustness must be evaluated individually for each architecture.

### E.7 MODEL ROBUSTNESS UNDER VARYING PERTURBATION BUDGETS

**Adversarial impact increases with attack budget, while targeted attacks exhibit saturation.** As shown in Figure 6, increasing the perturbation budget ($\epsilon$ or attack ratio $r$) leads to stronger degradation under untargeted and closer to targets in targeted settings. For untargeted attacks, especially under white-box conditions, high budgets result in substantial performance drops. For targeted attacks, however, we observe saturation: once the perturbation budget surpasses a certain threshold, further increases do not improve alignment with the target and may even reduce it due to oversteering.

**Model robustness is highly dataset-dependent.** Figure 7 presents robustness curves for each dataset under SimBA attacks, where we vary the perturbation budget $\epsilon$ and fix the attack ratio $r = 1$. For example, Moirai remains stable on US Births but degrades significantly on BizTObs-L2C, while TimesFM exhibits sharp performance drops across most datasets, indicating high vulnerability. In contrast, models like UniTS and Chronos show relatively moderate and consistent degradation. These results highlight the challenge of building TSFMs that are both accurate and robust across diverse real-world scenarios. Achieving such consistency remains a key open problem for safe and reliable deployment.

### E.8 FULL RESULTS OF TARGETED ATTACKS

**Global-targeted attacks (e.g., scaling or shifting the full forecast) are generally more effective than local-targeted attacks (e.g., modifying a subsegment).** Table 12 and Table 13 report RED$_{\text{NMAE}}$ scores under targeted attacks. This suggests that TSFMs often lack strong global constraints, making them susceptible to trajectory-wide manipulations. In contrast, localized targets

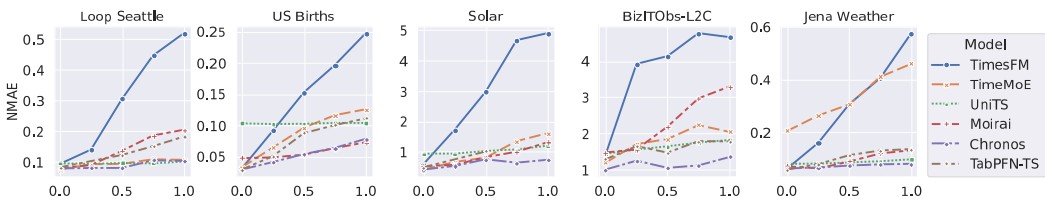

Figure 7: **Model robustness under SimBA attack across datasets.** We report the degradation in NMAE as the perturbation bound $\epsilon$ increases, with attack ratio $r = 1$.

are harder to exploit, likely due to inductive biases that enforce smoothness and temporal consistency—making it difficult to alter specific time steps without disrupting the overall sequence.

Table 12: **Targeted attacks (scaling and drifting).** We report the averaged $\text{RED}_{\text{NMAE}}$ across different attack budgets ($\epsilon \in \{0.25, 0.5, 0.75, 1\}$, $r \in \{0.25, 0.5, 0.75, 1\}$). Red denote the perturbed forecasts move closer to the target. Green denote predictions deviate further from the target.

| Model | PGD | | | | | SimBA | | | | |
|---|---|---|---|---|---|---|---|---|---|---|
| | $a = -1.0$ | $a = 0.5$ | $a = 2$ | $\beta = 0.05$ | $\beta = 0.1$ | $a = -1.0$ | $a = 0.5$ | $a = 2$ | $\beta = 0.05$ | $\beta = 0.1$ |
| TimesFM | 0.578 | 0.697 | 0.618 | 0.775 | 0.796 | 0.418 | 0.607 | 0.510 | 0.634 | 0.590 |
| TimeMoE | 0.225 | -0.131 | -0.021 | 0.145 | 0.124 | 0.416 | 0.599 | 0.379 | 0.538 | 0.460 |
| UniTS | 0.447 | -0.757 | -0.007 | -0.172 | 0.091 | 0.137 | -0.397 | -0.398 | -0.100 | 0.035 |
| Moirai | 0.475 | 0.595 | 0.323 | 0.604 | 0.637 | 0.178 | 0.159 | 0.021 | 0.294 | 0.251 |
| Chronos | - | - | - | - | - | 0.055 | -0.138 | -0.157 | 0.023 | 0.059 |
| TabPFN-TS | - | - | - | - | - | 0.365 | 0.688 | 0.476 | 0.484 | 0.486 |

Table 13: **Targeted attacks (local offset).** We report the Average $\text{RED}_{\text{NMAE}}$ across different attack budgets ($\epsilon \in \{0.25, 0.5, 0.75, 1\}$, $r \in \{0.25, 0.5, 0.75, 1\}$). We denote the perturbed region in the prediction horizon as $\langle \tau_{\text{start}}, \tau_{\text{end}} \rangle$, where $\tau \in [0, 1]$ is a normalized index ($\tau = 0$ corresponds to the first time step, and $\tau = 1$ to the last). Red highlights denote successful attacks where the perturbed forecasts move closer to the target. Green denote predictions deviate further from the target.

| Model | PGD | | | | SimBA | | | |
|---|---|---|---|---|---|---|---|---|
| | $\langle 0.75, 1 \rangle$ | $\langle 0, 0.25 \rangle$ | $\langle 0.5, 1 \rangle$ | $\langle 0, 0.5 \rangle$ | $\langle 0.75, 1 \rangle$ | $\langle 0, 0.25 \rangle$ | $\langle 0.5, 1 \rangle$ | $\langle 0, 0.5 \rangle$ |
| TimesFM | 0.294 | 0.265 | 0.465 | 0.433 | -0.100 | -0.099 | 0.193 | 0.194 |
| TimeMoE | -3.755 | -3.847 | -1.537 | -1.699 | -0.899 | -0.942 | -0.193 | -0.041 |
| UniTS | -6.162 | -5.984 | -2.610 | -2.511 | -5.066 | -4.611 | -2.260 | -2.044 |
| Moirai | 0.013 | 0.052 | 0.117 | 0.186 | -0.364 | -0.321 | -0.252 | -0.163 |
| Chronos | - | - | - | - | -0.611 | -0.648 | -0.492 | -0.403 |
| TabPFN-TS | - | - | - | - | -1.068 | -1.062 | -0.152 | -0.352 |

### E.9 MODEL PERFORMANCE UNDER DIFFERENT MODEL SIZE

**Larger models tend to be more vulnerable to gradient-based attacks.** As shown in Table 8, we observe a clear trend: models with larger parameter counts are generally more susceptible to gradient-based attacks. This may be due to the expanded capacity increasing the number of exploitable directions in the input space. An exception is TimesFM, where the 200M variant is more vulnerable than the 500M version. On the other hand, under black-box attacks (Appendix Table 8), model size does not show a consistent effect on robustness. These findings suggest that while scaling up model size can improve forecasting performance, it may also amplify vulnerability.

### E.10 ROBUSTNESS UNDER DIFFERENT PREDICTION HORIZONS

Figure 9 compares the robustness scores ($\text{RED}_{\text{NMAE}}$) of four TSFMs under PGD attacks across different prediction horizons. We observe that, in most cases, short-term forecasting is more susceptible to

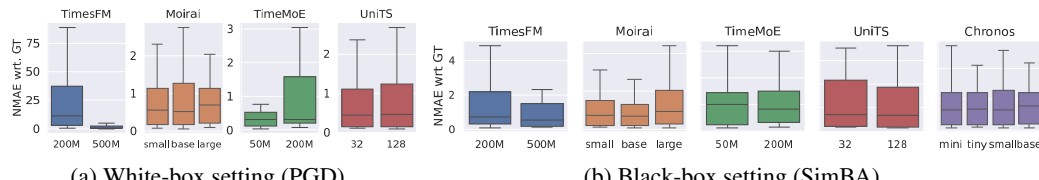

(a) White-box setting (PGD).    (b) Black-box setting (SimBA).

Figure 8: **Impact of model size on robustness under different attack strategies.** We evaluate TSFMs of varying scales under PGD and SimBA attacks, with fixed budget $\epsilon = 0.5, r = 1$.

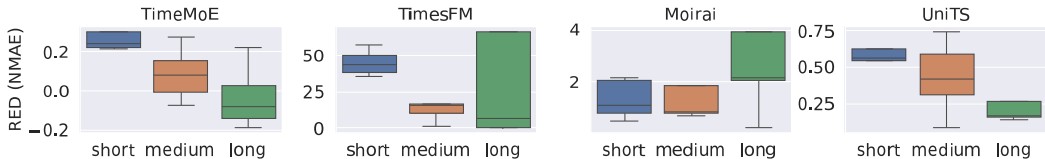

Figure 9: **Robustness under different prediction horizons.** RED$_{\text{NMAE}}$ of four TSFMs under PGD attacks with $\epsilon = 0.5$ and attack ratio $r = 1$. For short-term forecasting, the context length is set to 128; for medium- and long-term settings, it is 256.

adversarial perturbations. One possible explanation is that long-term forecasts are inherently less accurate, leading to lower baseline performance and therefore smaller relative error degradation.

### E.11 FULL RESULTS OF DEFENSE STRATEGY

Table 14: **Defense results on TimesFM (NMAE↓) of adversarial training.** *Clean*: natural error (no attack). *no def.*: vanilla model. *C-LAT*: cross-domain latent adversarial training. *C-IAT*: cross-domain input-space adversarial training. Both of them were fine-tuned on a KDD Cup 2018 dataset ($\epsilon = 0.5, r = 1$).

| Dataset | Clean | | | PGD | | | SimBA | | |
|---|---|---|---|---|---|---|---|---|---|
| | no def. | LAT | IAT | no def. | LAT | IAT | no def. | LAT | IAT |
| Loop Seattle | 0.113 | 0.099 | 0.105 | 1.213 | 0.154 | 0.170 | 0.167 | 0.128 | 0.141 |
| BizITObs-L2C | 2.904 | 2.385 | 2.238 | 19.950 | 5.085 | 4.092 | 6.495 | 3.860 | 3.281 |
| Electricity | 0.333 | 0.340 | 0.356 | 4.055 | 0.512 | 0.566 | 0.500 | 0.429 | 0.466 |
| ETT1 | 0.246 | 0.348 | 0.348 | 2.368 | 0.530 | 0.619 | 0.478 | 0.450 | 0.504 |
| Hier. Sales | 0.927 | 1.266 | 1.360 | 20.871 | 3.532 | 4.043 | 1.817 | 2.641 | 2.820 |
| Solar | 0.569 | 1.166 | 0.701 | 15.033 | 1.543 | 1.606 | 1.480 | 1.356 | 1.188 |
| US Births | 0.033 | 0.087 | 0.061 | 0.237 | 0.132 | 0.110 | 0.072 | 0.113 | 0.090 |

**Adversarial training provides the strongest robustness improvements.** Table 14 indicates that across nearly all datasets and under PGD, adversarial training substantially outperforms smoothing-based defenses. Both LAT and IAT significantly reduce worst-case errors, though LAT remains the stronger method overall. This demonstrates that adversarial fine-tuning, whether applied in latent or input space, can meaningfully mitigate gradient-based attacks. Importantly, adversarial training generally preserves clean accuracy, with LAT offering slightly smaller clean-error increases than IAT.

**Smoothing offers lightweight but unstable protection.** Inference-time smoothing continues to produce only modest and inconsistent improvements, as shown in Table 15. While it can reduce PGD errors (e.g., Electricity: $4.055 \rightarrow 1.654$ at $K = 7$), its protection is consistently weaker than both LAT and IAT. Additionally, smoothing often harms clean accuracy, especially on low-noise, highly structured datasets (e.g., Solar: $0.569 \rightarrow 0.925$ at $K = 7$), reinforcing its unfavorable robustness–utility trade-off.

**Dataset-specific trade-offs remain challenging.** Defense behavior varies considerably across datasets. In certain domains, adversarial training may even degrade robustness: on Hier. Sales

Table 15: **Defense results on TimesFM (NMAE↓) of input smoothing.** *Clean*: natural error (no attack). *no def.*: vanilla model. *(K=3/5/7)*: inference-time moving-average smoothing with kernel size $K$.

| Dataset | Clean | | | | PGD | | | | SimBA | | | |
|---|---|---|---|---|---|---|---|---|---|---|---|---|
| | no def. | (K=3) | (K=5) | (K=7) | no def. | (K=3) | (K=5) | (K=7) | no def. | (K=3) | (K=5) | (K=7) |
| Loop Seattle | 0.113 | 0.106 | 0.108 | 0.112 | 1.213 | 0.416 | 0.399 | 0.308 | 0.167 | 0.154 | 0.121 | 0.121 |
| BizITObs-L2C | 2.904 | 2.822 | 2.887 | 2.959 | 19.947 | 14.325 | 8.331 | 9.149 | 6.495 | 3.853 | 3.967 | 3.076 |
| Electricity | 0.333 | 0.333 | 0.346 | 0.345 | 4.055 | 2.227 | 1.698 | 1.654 | 0.500 | 0.420 | 0.457 | 0.462 |
| ETT1 | 0.246 | 0.267 | 0.311 | 0.352 | 2.368 | 1.364 | 1.011 | 0.903 | 0.478 | 0.407 | 0.438 | 0.416 |
| Hier. Sale | 0.927 | 1.199 | 1.228 | 1.686 | 20.871 | 6.290 | 8.492 | 4.683 | 1.817 | 2.795 | 3.288 | 1.377 |
| Solar | 0.569 | 0.650 | 0.795 | 0.925 | 15.033 | 6.002 | 4.177 | 3.299 | 1.480 | 1.593 | 1.222 | 1.171 |
| US Birth | 0.033 | 0.083 | 0.104 | 0.100 | 0.237 | 0.205 | 0.301 | 0.358 | 0.072 | 0.105 | 0.111 | 0.117 |

under SimBA, IAT increases error from $1.817 \rightarrow 2.820$, and LAT from $1.817 \rightarrow 2.641$. Similarly, smoothing amplifies errors in several cases. These failures suggest that misalignment between the defense objective and dataset/attack structure can cause regressions, and that no single defense universally dominates across time-series regimes.

### E.12 COMPARISON TO TRADITIONAL FORECASTING MODELS

Table 16: Clean forecasting performance (NMAE↓) of TSFMs and supervised baselines on long-term datasets (ETT, Weather, context and prediction length 96) and short-term datasets (Exchange, Electricity, context and prediction length 24). . Supervised models are trained on each dataset.

| Dataset | **TSFMs** | | | **Supervised** | | |
|---|---|---|---|---|---|---|
| | TimesFM | Moirai | UniTS | GRU | TCN | Informer |
| ETTh1 | **0.3313** | 0.3425 | 0.4080 | 0.3874 | 0.4749 | 0.4985 |
| ETTh2 | **0.1954** | 0.2019 | 0.1975 | 0.2036 | 0.2107 | 0.2441 |
| ETTm1 | **0.3281** | 0.5163 | 0.4419 | 0.3496 | 0.3815 | 0.3572 |
| ETTm2 | **0.1596** | 0.1655 | 0.1729 | 0.1668 | 0.1727 | 0.2391 |
| Weather | **0.0789** | 0.3670 | 0.0994 | 0.7746 | 0.2182 | 0.5139 |
| Exchange | **0.0244** | 0.0252 | 0.0295 | 0.0295 | 0.0416 | 0.0902 |
| Electricity | 0.3445 | 0.3582 | 0.3638 | 0.8305 | **0.1275** | 0.7845 |

**Model configurations.** For Informer, we use a standard encoder–decoder setup with model dimension 512 and 8 attention heads. The network has 2 encoder layers and 1 decoder layers, fixed embeddings, and dropout 0.1. The GRU forecaster is a simple multi-step model with hidden size 64, two recurrent layers, and dropout 0.1. It takes the same context windows as input and outputs quantile forecasts for the prediction horizon. The TCN forecaster is a temporal convolutional network with three convolutional blocks, kernel size 3, and dropout 0.2. As with GRU, it operates on the same context windows and produces quantile forecasts over the horizon.

**Analysis.** The clean results show that TSFMs already provide strong zero shot performance on most datasets and often outperform simple supervised models trained from scratch. The ordering under clean performance does not match the ordering under adversarial robustness. Models that perform best in terms of NMAE are not necessarily the most robust when subjected to PGD attacks. This supports our main observation that clean accuracy and adversarial robustness are only weakly aligned for both TSFMs and conventional models, and that robustness must be evaluated explicitly with a dedicated threat model rather than inferred from clean metrics alone.

