# OpenReview forum: "Are Time-Series Foundation Models Deployment-Ready? A Systematic Study of Adversarial Robustness Across Domains"
_ICLR.cc/2026/Conference — Submitted to ICLR 2026_

### Official Review · Reviewer_1xNi · 2025-10-27

**Soundness:** 1
**Presentation:** 2
**Contribution:** 2
**Rating:** 4
**Confidence:** 4

**Summary:**

This paper presents a comprehensive evaluation of the adversarial robustness of Time-Series Foundation Models (TSFMs). Despite their growing adoption in high-stakes applications, the security of TSFMs under adversarial conditions remains underexplored. This work addresses that gap through a unified framework that evaluates various threat models, attack methods, and robustness metrics.

**Strengths:**

- The paper addresses the underexplored adversarial robustness of TSFMs, which are increasingly used in critical real-world applications.
- The study presents a unified framework covering diverse threat models, attack types, and metrics, offering a broad and systematic analysis of TSFM vulnerabilities.
- The evaluation provides actionable insights for model selection and defense in adversarial settings.

**Weaknesses:**

- **Outdated Attack and Defense Techniques**:
  Although the evaluation spans different goals, capabilities, and knowledge levels, the choice of attack and defense methods appears outdated and insufficient. The paper still relies on traditional computer vision attacks, with the most recent being SimBA from 2019. Similarly, the defense strategies used are no longer state-of-the-art. I suggest incorporating more recent adversarial attacks and defenses specifically tailored to time-series data.

- **Lack of Time-Series–Specific Insights and Misleading Claims**:
  The results echo patterns already well-documented in the computer vision literature, with few observations that highlight the unique characteristics of time-series forecasting. Some statements also appear overstated or even potentially incorrect:

  1. For instance, the claim “Failures are model-specific, with limited cross-model transfer” is not surprising, especially since the transfer attacks are seemingly conducted by directly applying white-box perturbations to other models, without any established transfer-enhancing strategies. The paper misses the opportunity to explore whether classic transferability techniques are effective for time-series models.
  2. Additionally, using the same $\epsilon$ across datasets is problematic because time-series data can have vastly different scales. This makes cross-dataset comparisons unfair and weakens the conclusion that certain datasets are more vulnerable. A per-dataset normalization or scale-aware perturbation constraint would be more appropriate.
- **Ambiguous Writing and Visualization**:
  Several parts of the paper are unclear or confusing, making it difficult to assess the actual threat. For example, in **Figure 1**, key parameters (like $\epsilon$) are not specified. The perturbation visually appears to be much larger than $\epsilon$ = 1 used in **Table 2**, suggesting inconsistencies between figures and main experiments. **Figure 3** is similarly vague—only a range of [0.25, 5] is mentioned for the budget, but no specific $\epsilon$ values are plotted. I recommend the authors improve the clarity and specificity of all experimental settings and avoid vague or potentially misleading descriptions.

**Questions:**

- What is q in Figures 1 and 3? This variable is never defined in the main text, which leaves the reader confused.

- It is unclear whether the values of $\epsilon$ {0.25, 0.5, 0.75, 1} and r {0.25, 0.5, 0.75, 1} in Table 2 are paired (i.e., (0.25, 0.25), (0.5, 0.5), etc.) or fully cross-matched. If the latter, then averaging all results may obscure the effect of low perturbation regimes, since the strongest ($\epsilon$ = 1, r = 1) combinations dominate the average. Please consider presenting disaggregated results for different perturbation levels to clarify how performance degrades under small perturbations.

- What is the perceptual or practical impact of different perturbation magnitudes? Since Figure 1 and Figure 3 lack explicit parameter annotations, the perturbations—especially in Figure 3(b)—appear visually large (perhaps >1000 in value). I suggest incorporating explicit metrics for perturbation imperceptibility to better support claims of realistic risk.

- The phrase “the maximum change per step” is likely incorrect, since the method appears to apply a single-step change. Please remove “per step” for accuracy.

- The current black-box attack is purely query-based. Have the authors considered traditional transfer-enhancing methods (e.g., input diversity, ensemble gradients, surrogate fine-tuning)? It would be valuable to assess whether such techniques, commonly used in vision, also work for time-series models.

- The true risk posed by these attacks remains unclear. While the RED score may indicate high vulnerability, the robustness curves suggest that small perturbations cause minimal absolute error. Would such small distortions result in meaningful or observable consequences in downstream applications? The paper would benefit from a clearer discussion of what constitutes a “successful” or threatening perturbation in real-world settings.

---

> ### Author Response · Authors · 2025-11-14
> **Response to Reviewer 1xNi (1/3)**
>
> Thank you for the careful and constructive review. We sincerely appreciate the opportunity to clarify the novelty, scope, and contributions of our work.
>
> ## W1. Outdated Attack and Defense Techniques
>
> We appreciate the reviewer’s concern and would like to clarify the deliberate design behind our methodological choices.
>
> **Our Core Goal is Diagnosis and Guidance, Not New SOTA Methods**
>
> Our core goal is diagnosis and practical guidance, rather than algorithmic invention. We are not trying to prove that “some novel attack can break TSFMs”—we agree this is almost a given.
>
> Instead, our research question is: “Can these new, large-scale TSFMs withstand the most basic, well-known attacks?” This question is crucial for deployment and is best answered using standard, model-agnostic baselines such as PGD and SimBA, rather than highly specialized attacks. Our defenses follow the same philosophy: we study minimal, deployment-feasible interventions to see whether existing checkpoints can be strengthened without redesigning and minimal finetuning.
>
> **Alarming Efficacy of “Outdated” Attacks**
>
> We respectfully disagree with the assessment that these attacks are “insufficient.” As shown in Table 2, these so-called traditional attacks are in fact alarmingly effective: a standard PGD attack can cause more than a 40× increase in error on TimesFM.
> If a large, expensive foundation model pretrained on massive time-series corpora can be catastrophically degraded by a 2019 black-box attack and an even older white-box method, this points to a deep, structural vulnerability rather than a weakness of the attack suite.
> Conversely, if only complex, newly designed, time-series–specific attacks could break TSFMs, that would suggest these models are relatively robust. Our results show instead that **the bar for a successful attack is strikingly low**, which we believe is the more urgent and important finding.
>
> **Time-series–specific Customizations**
>
> While the core algorithms are classical, the framework is **explicitly tailored to time series**:
>
> 1. Threat Model (Section 3.2): Our entire threat model, including the variance-normalized perturbation budget ($\epsilon^*$) and sparse-ratio attacks ($r$), was designed specifically for time-series.
> 2. Attack Basis (Appx. B.1): We explore a Wavelet-based SimBA variant, explicitly leveraging time-frequency structure to better align with time-series characteristics.
> 3. Robustness Metric (Section 3.4): Our proposed RED metric is scale-invariant and designed to (i) compare robustness across heterogeneous datasets and models, and (ii) capture both deviation (untargeted) and target-alignment (targeted) in a unified way.
>
> **“Outdated” Techniques Led to “Brand New” Insights**
>
> Using classical attack/defense on TSFMs reveals several new, time-series–specific robustness phenomena:
>
> - Context-length sensitivity: longer look-back windows, while boosting accuracy, can act as an amplifier for attacks.
> - Latent-space generalization: adversarial fine-tuning on one dataset substantially improves robustness on unseen domains, which is not obvious in a modality without shared semantics.
> - Structure-preserving targeted control: adversaries can insert desired trends or seasonal distortions while keeping the input signal visually plausible—behavior unique to time series.
>
> Even if some conclusions may appear intuitive in hindsight, they have not previously been demonstrated for TSFMs. Our work provides systematic empirical evidence.
>
> In summary, our work shows that for TSFMs, we don't even need the SOTA arsenal. Basic tools are enough to cause catastrophic failure. We believe this finding is far more alarming and significant than simply breaking a new model with a new attack.

---

> ### Author Response · Authors · 2025-11-14
> **Response to Reviewer 1xNi (2/3)**
>
> ## W2. Misleading Claims
>
> > the claim “Failures are model-specific, with limited cross-model transfer” is not surprising, …. The paper misses the opportunity to explore whether classic transferability techniques are effective for time-series models.
> >
>
> In CV, transfer-enhancing techniques are powerful because they build on a strong baseline of transferability. For TSFMs, our empirical observations suggest a very different picture.
>
> Beyond the experiments reported in the main paper, we conducted additional, off-paper tests with:
>
> - momentum iterative methods,
> - several forms of universal adversarial perturbations (sample-level, batch-level, with momentum and Adam),
> - model-based adaptive perturbation generators.
>
> Across these attempts, we consistently observed weak or negligible cross-model transfer among TSFMs. This indicates that current vulnerabilities are likely architecture-bound rather than shared across models via common feature spaces.
>
> In a regime where baseline transfer is almost absent, applying CV-style “transfer-enhancing” techniques is not well motivated—they attempt to amplify a property that the domain does not yet clearly exhibit. We will tighten our wording in the paper to make clear that our “limited transfer” conclusion is based on current available evidence, and to avoid overstating this claim.
>
> > using the same across datasets is problematic because time-series data can have vastly different scales. …. A per-dataset normalization or scale-aware perturbation constraint would be more appropriate.
> >
>
> We **fully agree** with the reviewer's principle: using a same absolute $\epsilon$ across datasets with very different scales would indeed be unfair and could invalidate cross-dataset comparisons.
>
> Crucially, this is **not** what we do.
>
> - In Section 3.2 (Lines 213–215) we explicitly define a variance-normalized budget $\epsilon^{*} = \epsilon \cdot \text{var}(x)$, where all $\epsilon$ values reported in our paper (e.g., 0.25, 0.5) are therefore relative, normalized budgets, not absolute values.
> - All TSFMs we evaluated heavily rely on internal normalization (ReVIN, Z-score, etc.), meaning they operate on normalized latent spaces, not raw-scale inputs (Appx. A).
> - All evaluation metrics (NMAE, RED, etc.) are scale-invariant (Section 3.4, App. B.2).
>
> Together, these design choices ensure that our perturbations and robustness measurements are **scale-invariant** and that cross-dataset comparisons are fair. We will highlight this more explicitly in the main text.
>
> ## W3. Ambiguity in Writing and Visualization
>
> We thank the reviewer for calling out clarity issues in the figures and agree they should be improved.
>
> We acknowledge that some visualizations were not described with sufficient parameter detail, which may cause confusion. We will revise these figures to ensure all parameters, scales, and perturbation budgets are precisely reported.
>
> - Figure 1: The perturbation may appear larger because the y-axis is auto-scaled by Matplotlib to maximize visibility of small deviations. In both figures, perturbation magnitude is bounded in $\epsilon \leq 1$. We will label $\epsilon$ explicitly.
> - Figure 3: We apologize for the vague presentation. We will annotate the exact budgets used in each subfigure and make the figure self-contained.
>
> ## Response to Questions
>
> > Q1. What is q in Figures 1 and 3? This variable is never defined in the main text, which leaves the reader confused.
> >
>
> In both figures, q denotes the **forecast quantile** output by TSFMs.
>
> - q = 0.5 corresponds to the median prediction
> - q = 0.1 and q = 0.9 define the lower and upper uncertainty bounds
>
> We will define this clearly in the main text and captions.
>
> > Q2. It is unclear whether the values of $\epsilon \in \{0.25, 0.5, 0.75, 1.0\}$ and $r \in \{0.25, 0.5, 0.75, 1.0\}$ in Table 2 are paired (i.e., (0.25, 0.25), (0.5, 0.5), etc.) or fully cross-matched. If the latter, then averaging all results may obscure the effect of low perturbation regimes, since the strongest ($\epsilon = 1$, $r = 1$) combinations dominate the average. Please consider presenting disaggregated results for different perturbation levels to clarify how performance degrades under small perturbations.
> >
>
> The values $\epsilon$ and $r$ are fully cross-matched, not paired. Each attack is therefore evaluated under all $4 \times 4 = 16$ combinations of perturbation budgets.
>
> We agree that averages may obscure low-budget behavior. To address this, **we already include disaggregated robustness curves across different perturbation levels in Appendix E.7 (Figures 6 and 7).** These plots separately visualize the effect of varying $\epsilon$ and $r$. We will add a clear pointer to this in the main text.

---

> > ### Author Response · Authors · 2025-11-14
> > **Response to Reviewer 1xNi (3/3)**
> >
> > > Q3. What is the perceptual or practical impact of different perturbation magnitudes? Since Figure 1 and Figure 3 lack explicit parameter annotations, the perturbations—especially in Figure 3(b)—appear visually large (perhaps >1000 in value). I suggest incorporating explicit metrics for perturbation imperceptibility to better support claims of realistic risk.
> > >
> >
> > We appreciate this important question. Unlike images and text, time series lack a universally accepted notion of human “imperceptibility.” In practice, even visibly noticeable perturbations can be realistic because **sensor noise, calibration drift, and seasonal effects** naturally produce structured deviations that resemble our perturbations. In real deployment, the clean signal is not observable, and perturbations can be easily masked by system noise. As discussed in the paper (Table 7), the attacked inputs closely resemble valid trajectories, and simple statistical checks often fail to distinguish them.
> >
> > Importantly, relative magnitude determines perceptual realism in time series. All perturbations in our study are generated using scale-aware constraints, ensuring that the injected deviations remain modest compared with the dataset’s dynamic range—even when the absolute values look large on high-amplitude series. In Figure 3(b), perturbations appear large primarily because some datasets have large raw amplitudes. The **relative** perturbation always satisfies our constrained budget $\epsilon \le 1.0$ (variance-normalized).
> >
> > > Q4. The phrase “the maximum change per step” is likely incorrect, since the method appears to apply a single-step change. Please remove “per step” for accuracy.
> > >
> >
> > Thank you for pointing out this potential ambiguity. In Eq. (4),
> >
> > $S = \{\delta \in \mathbb{R}^L : \|\delta\|0 \le rL,\ \|\delta\|\infty \le \epsilon\}$,
> >
> > $\epsilon$ bounds the **per–time-step** change, i.e., $|\delta_t| \le \epsilon$ for each time index t. Our phrase “maximum change per step” referred to this per–time-step bound, not an optimization step.
> >
> > We agree that this wording can be confusing and may suggest a single update step. In the revision, we will replace it with a more precise description.
> >
> > > Q5. The current black-box attack is purely query-based. Have the authors considered traditional transfer-enhancing methods (e.g., input diversity, ensemble gradients, surrogate fine-tuning)? It would be valuable to assess whether such techniques, commonly used in vision, also work for time-series models.
> > >
> >
> > We agree that exploring transfer-enhancing strategies is interesting. However, as we mentioned in W2, they presuppose non-trivial baseline transferability, which our preliminary experiments did not observe.
> >
> > Moreover, many classic transfer tricks (e.g., input diversity) rely on semantics-preserving transformations (crops, resizes) that have no direct analogue in time series: rescaling, warping, or jittering often destroy the temporal meaning of the signal and break the forecasting task.
> >
> > Given this, we chose to focus on query-based black-box attacks (SimBA/ZOO-like), which directly match realistic threats to deployed TSFMs (no gradients, no internal access, no surrogates).
> >
> > We will clarify this rationale in the revision and will release all attack code to encourage further exploration of transfer-based methods in future work.
> >
> > > Q6. The true risk posed by these attacks remains unclear. While the RED score may indicate high vulnerability, the robustness curves suggest that small perturbations cause minimal absolute error. Would such small distortions result in meaningful or observable consequences in downstream applications? The paper would benefit from a clearer discussion of what constitutes a “successful” or threatening perturbation in real-world settings.
> > >
> >
> > We fully agree that a successful attack must have a real-world impact, not just a statistical deviation. **The key risk we demonstrate is behavioral control of forecasts, not the magnitude of the error itself.**
> >
> > Targeted attackers do not aim to maximize error; they aim to shape the model’s forecast toward a malicious pattern:
> >
> > - In Figure 3(c), a drift attack subtly but consistently pushes forecasts upward.
> > - In Figure 1(a), modifying a few historical points yields controlled changes at selected forecast horizons.
> >
> > Even if per-step absolute errors are small, these controlled distortions can severely affect downstream systems, which often react to **trends, turning points, or sustained biases**, not just pointwise error.
> >
> > This is exactly **why we introduce RED**: in targeted attacks, **success** means *making the model’s forecast close to an attacker-defined target*, not far from the clean forecast. A high RED indicates the attacker has effectively *steered* the model’s behavior, which is both practically and operationally dangerous. RED thus captures the threat more faithfully than NMAE or RMSE in scenarios where the attacker’s goal is stealthy behavioral influence.

---

### Official Review · Reviewer_iwVq · 2025-10-31

**Soundness:** 2
**Presentation:** 3
**Contribution:** 2
**Rating:** 2
**Confidence:** 5

**Summary:**

The paper presents an empirical study on the adversarial robustness of multiple Time-Series Foundation Models (TSFMs)—TimesFM, TimeMoE, UniTS, Moirai, Chronos, and TabPFN-TS—under white-/black-box and targeted/untargeted attacks.
It proposes a unified evaluation framework with the Relative Error Deviation (REDE) metric, a mixed-norm perturbation budget (ℓ₀ + ℓ∞), and two defenses (inference-time smoothing and latent adversarial training LAT).
Experiments cover six models and eight datasets, showing that minor perturbations can cause large forecast errors and that LAT improves worst-case robustness.

**Strengths:**

1. The unified threat model (goal / capability / knowledge) and hybrid-norm constraint are technically well-defined and well-motivated for time-series data.

2. The work systematically examines model-specific failure modes, horizon sensitivity, and context-length effects.

3. Defense results are quantitatively compelling. In-domain LAT improves worst-case NMAE up to 10× under PGD and generalizes well out-of-domain.

4. Reproducibility statement and released code enhance reliability.

**Weaknesses:**

**1. Over-claimed novelty and limited contribution boundary**

The paper repeatedly claims to be **“the first large-scale, systematic robustness evaluation of TSFMs.”**
However, two peer-reviewed works have already addressed adversarial robustness of TSFMs directly:

**Adversarial Vulnerabilities in Large Language Models for Time Series Forecasting - AISTATS 2025**
Performs a systematic, cross-model and cross-dataset robustness analysis including TSFM such as TimeGPT, demonstrating that small, structured perturbations cause significant and controllable forecast distortions.

**Temporally Sparse Attack for Fooling LLMs in Time Series Forecasting - ICML 2025 workshop**
Introduces a cardinality-constrained optimization attack that manipulates only ≈ 10 % of time steps while severely degrading forecasts of LLM-based TSFMs (including TimeGPT), directly exposing their adversarial weaknesses.

Because both prior studies explicitly involve TSFMs and directly reveal their adversarial vulnerabilities, the core finding of this submission is no longer novel.
The main difference lies in the model family extension (from LLM-TSFMs to other pretrained forecasting models), which constitutes an incremental replication, not a conceptual breakthrough.
Consequently, the repeated “first large-scale, systematic robustness evaluation of TSFMs” statements appear over-claimed and should be toned down to avoid misleading readers.

**2. Limited methodological innovation**

The two proposed defenses—moving-average smoothing and LAT—are straightforward adaptations of known techniques with modest empirical tuning.
No comparison is provided against input-space adversarial training, noise-based defenses, or detection mechanisms, limiting methodological novelty.

**3. Evaluation gaps**

The attack suite omits universal or adaptive baselines (e.g., AutoAttack-style ensembles).

No comparison to traditional forecasting models (LSTM, TCN, Informer), leaving unclear whether TSFMs are uniquely fragile or simply representative of general deep forecasting vulnerabilities.

Mechanistic insights remain descriptive: while horizon-boundary and context-length sensitivities are observed, there are no controlled ablations (e.g., disabling patchification or varying decoder type) to establish causality.

**Questions:**

1. Compare LAT with input-space adversarial fine-tuning under equal training cost and report clean-accuracy trade-offs.

2. Have you tested universal or adaptive attacks? It will be intresting to see these attacks' performance on TSFM.

3. Can you conduct controlled ablations (e.g., without patchification or using alternative decoding heads) to validate causal explanations for observed vulnerabilities?

4. Add non-foundation baselines to clarify whether fragility is specific to TSFMs.

---

> ### Author Response · Authors · 2025-11-14
> **Response to Reviewer iwVq (1/2)**
>
> Thank you for the careful and constructive review. We sincerely appreciate the opportunity to clarify the novelty, scope, and contributions of our work.
>
> ## Novelty and Relation to Prior Work
>
> We appreciate the reviewer highlighting the two recent papers on *LLM-based* time-series forecasting robustness. These works analyze the adversarial behavior of **LLMs adapted for forecasting**, such as TimeGPT, using threat models, tokenization, and decoding pipelines tailored to LLMs. Our paper focuses on a fundamentally different, complementary model family: **non-LLM, single-modal pretrained TSFMs** (e.g., TimesFM, TimeMoE, Moirai, Chronos).
>
> These models differ from LLM-based systems in:
>
> - architecture design
> - pretraining corpora and objectives,
> - inference pipelines and input semantics,
> - zero-shot forecasting interfaces.
>
> To our knowledge, **no prior work has conducted a systematic robustness evaluation of this non-LLM TSFM family, in zero-shot, frozen-checkpoint settings**, nor documented their shared failure modes, domain-transfer vulnerabilities, or cross-domain adversarial fine-tuning behavior.
>
> We will revise the wording to accurately describe our contribution and cite the LLM-based robustness studies as complementary.
>
> ## Contribution Boundary
>
> Our goal is diagnostic rather than algorithmic, i.e., to provide the community with the first unified, time-series–specific threat model and evaluation protocol tailored to TSFMs.
>
> While adversarial risk is generally known in ML, its **manifestation in modern TSFMs, which pretrained across domains and used zero-shot, was previously unknown**. Our contributions include:
>
> - establishing a **TS-specific robustness framework** (structured targets, scale-aware budgets, horizon-aligned perturbations);
> - evaluating robustness across multiple SOTA TSFMs under a common protocol;
> - revealing **consistent, non-obvious vulnerability patterns** that hold across architectures and datasets.
>
> Though our defenses are adaptations of known ideas, they uncover **new, TSFM-specific behaviors.** For example, cross-domain latent adversarial tuning that fine-tuning on an unrelated dataset can achieve 80–95% of robustness gains on a completely out-of-domain dataset (Table 3), a non-obvious property unique to TSFMs and valuable for practitioners.
>
> ## Methodological Innovation and Alternative Defenses
>
> Our study focuses on **minimally invasive, model-agnostic defenses** that apply to frozen, zero-shot TSFMs without requiring architectural access and minimal retraining.
>
> Heavy noise defenses, full-model retraining, or complex detectors often degrade clean accuracy and require in-domain data, which contradicts the deployment regime of TSFMs.
>
> To address the reviewer’s suggestion, we added a comparison with input-space adversarial training (IAT) under equal training cost. The results (included in the response) show that across datasets, both LAT and IAT serve as effective defenses against strong white-box PGD attacks. LAT outperforms IAT on five out of seven datasets, indicating that LAT provides more stable robustness gains under PGD. In contrast, IAT tends to induce a slightly smaller degradation in clean accuracy. Their performance against SimBA is mixed, with neither method consistently dominating, and both can degrade robustness on some datasets, which highlights the difficulty of defending TSFMs in general.
>
> Table. Defense results on TimesFM (NMAE↓). Cross-domain LAT vs. Cross-domain Input-space adversarial training (IAT), both of which fine-tuned on a KDD Cup 2018 dataset (eps=0.5, r=1).
>
> | Dataset | Clean - no def | Clean - LAT | Clean - IAT | PGD - no def | PGD - LAT | PGD - IAT | SimBA - no def | SimBA - LAT | SimBA - IAT |
> | --- | --- | --- | --- | --- | --- | --- | --- | --- | --- |
> | Loop Seattle | 0.113 | 0.099 | 0.105 | 1.213 | 0.154 | 0.170 | 0.167 | 0.128 | 0.141 |
> | BizITObs-L2C | 2.904 | 2.385 | 2.238 | 19.95 | 5.085 | 4.092 | 6.495 | 3.860 | 3.281 |
> | Electricity | 0.333 | 0.340 | 0.356 | 4.055 | 0.512 | 0.566 | 0.500 | 0.429 | 0.466 |
> | ETT1 | 0.246 | 0.348 | 0.348 | 2.368 | 0.530 | 0.619 | 0.478 | 0.450 | 0.504 |
> | Hier. Sales | 0.927 | 1.266 | 1.360 | 20.871 | 3.532  | 4.043 | 1.817 | 2.641 | 2.820 |
> | Solar | 0.569 | 1.166 | 0.701 | 15.033 | 1.543 | 1.606 | 1.480 | 1.356 | 1.188 |
> | US Births | 0.033 | 0.087 | 0.061 | 0.237 | 0.132 | 0.110 | 0.072 | 0.113 | 0.090 |

---

> ### Author Response · Authors · 2025-11-14
> **Response to Reviewer iwVq (2/2)**
>
> ## On Universal or Adaptive Attack Baselines
>
> Our attack suite focuses on deployment-grounded, semantically meaningful perturbations (global scale shifts, drift, local offsets, sign flips), which act as universal attacks for time-series systems. Designing the strongest possible attack is not the objective of this work; rather, we aim to stress-test TSFMs under realistic perturbations aligned with time-series semantics.
>
> Universal or adaptive ensembles (e.g., AutoAttack-style pipelines) are interesting extensions but lie outside the present study’s emphasis on interpretable, operationally plausible perturbations. We will clarify this in the revision.
>
> ## Comparison to traditional forecasting models
>
> We have added experiments  for GRU, TCN, and Informer, each trained on the target dataset under standard supervised settings. All models are tested using the same PGD untargeted attack ($\epsilon=0.5, r=0.5$) and the $\text{RED}_{\text{NMAE}}$ robustness metric.
> The new results show that:
> - Traditional supervised models are generally more robust than zero-shot TSFMs under the same perturbation budget, often by a substantial margin.
> - Within conventional models, architectural complexity correlates with reduced robustness: Informer is notably less robust than GRU and TCN, consistent with prior findings that high-dimensional, more linearized models tend to be more vulnerable.
> - The vulnerability patterns of TSFMs differ qualitatively from those of traditional models, reinforcing that robustness in pretrained, cross-domain TSFMs is governed by mechanisms not captured in conventional supervised settings.
>
> These findings reinforce that our aim is not to compare “which model class is more fragile,” but to **characterize robustness behaviors intrinsic to TSFMs** as a new pretraining-based forecasting paradigm. Conventional models serve as a useful reference point, but their robustness is governed by different factors and does not explain the TSFM-specific failure modes our study aims to uncover.
>
> We appreciate the reviewer’s valuable suggestion and have incorporated the comparison into the revision. Our evaluation framework is model-agnostic, and we will release code so that additional architectures can be assessed under the same threat model.
>
> Table. Robustness comparison between TSFMs and conventional forecasting models. We report $RED_{\text{NMAE}}$ under PGD untargeted attacks with $\epsilon=0.5, r=0.5$. Conventional forecasting models (GRU, TCN, and Informer) are trained specifically for each dataset. Lower $\text{RED}_{\text{NMAE}}$ indicates better robustness.
>
> |  | TimsFM | Moirai | UniTS | GRU | TCN | Informer |
> | --- | --- | --- | --- | --- | --- | --- |
> | ETTh1 | 2.1038 | 0.6814 | 0.0346 | 0.1456 | 0.0510 | 0.1302 |
> | ETTh2 | 3.3649 | 0.5909 | 0.0587 | 0.1203 | 0.0043 | 0.0344 |
> | ETTm1 | 6.5620 | 1.0180 | 0.1208 | 0.1087 | 0.0522 | 0.2252 |
> | ETTm2 | 5.2638 | 1.0133 | 0.0798 | 0.1060 | 0.0023 | 0.1376 |
> | Exchange | 4.2828 | 0.5000 | 0.0271 | 0.0537 | 0.0072 | 2.0443 |
> | Weather | 5.2788 | 0.4640 | 0.1660 | 0.0422 | 0.0050 | 0.3578 |
> | Electricity | 2.1000 | 0.6019 | 0.4733 | 0.0011 | 0.2313 | 0.0212 |
>
> ## Mechanistic Ablations
>
> We agree that deeper mechanistic studies (e.g., removing patchification, varying decoders) would be valuable. However, TSFMs are released as fixed pretrained checkpoints. Altering components such as patching, tokenization, or decoder architecture would produce entirely new models, confounding architecture with pretraining distribution, objective, and scaling.
>
> Our objective is therefore not to assert causal mechanisms but to provide a systematic, deployment-aligned characterization of robustness behaviors that consistently appear across multiple TSFMs with different architectures. These cross-model regularities are precisely what motivates deeper mechanistic exploration in future work. We will clarify this scope and explicitly position the mechanistic probes you suggested as important follow-up directions that require model-level retraining rather than ablation.
>
> ---
>
> Thank you again for the constructive comments. We will revise the manuscript to:
>
> - accurately position our contribution relative to LLM-based work,
> - explicitly narrow the claim regarding novelty,
> - clarify the diagnostic scope,
> - incorporate the suggested comparisons and additional explanation where appropriate.

---

### Official Review · Reviewer_ib3J · 2025-11-01

**Soundness:** 3
**Presentation:** 4
**Contribution:** 2
**Rating:** 4
**Confidence:** 5

**Summary:**

This submission presents an adversarial attack framework for zero-shot time series forecasting models, designed to assess the robustness of time-series foundation models. Two types of attacks are introduced: a white-box attack based on the Fast Gradient Sign Method (FGSM) and a black-box attack employing zero-order optimization. To improve model robustness, the study further proposes two defense strategies: a filter-based preprocessing defense and an adversarial training–based defense.

**Strengths:**

The research topic is important and timely. The vulnerability and robustness of time series foundation models remain underexplored.

The experimental design is comprehensive. Six representative models are evaluated across eight diverse datasets, providing convincing evidence to support the study’s findings.

**Weaknesses:**

The primary weakness of this submission lies in its limited technical novelty. The manuscript mainly applies existing adversarial attack and defense techniques to zero-shot time series forecasting models, without introducing fundamentally new methodologies. Consequently, the proposed attacks can largely be mitigated by existing defense mechanisms.

More specifically:

1. **Relation to Prior Work**: The submission does not clearly articulate its relationship or distinction from prior studies. For example:
  [1] introduced FGSM-based white-box attacks and proposed filter-based and adversarial fine-tuning defenses for time series forecasting;[2] presented a zero-order optimization (SPSA)-based black-box attack for forecasting models; and [3] developed targeted, gradient-free black-box attacks specifically for zero-shot, LLM-based time series forecasting models.

2. **Novelty and Significance**: The reliance on established adversarial attack and defense methods reduces the overall contribution of the paper. The presented framework does not demonstrate a substantial methodological advancement beyond existing literature.

3. **Insights on Foundation Models**: The work does not sufficiently uncover new challenges or unique vulnerability patterns specific to time series foundation models under adversarial conditions.

4. **Comparative Robustness Analysis**: A direct robustness comparison between zero-shot time series foundation models and conventional forecasting models is missing. Including such baselines would strengthen the evaluation and clarify whether foundation models exhibit distinct robustness characteristics.

**Reference**

[1] Liu, Linbo, et al. "Robust Multivariate Time-Series Forecasting: Adversarial Attacks and Defense Mechanisms." ICLR (2023).

[2] Zhu, Lyuyi, et al. "Adversarial diffusion attacks on graph-based traffic prediction models." IEEE Internet of Things Journal 11.1 (2023): 1481-1495.

[3] Liu, Fuqiang, et al. "Adversarial Vulnerabilities in Large Language Models for Time Series Forecasting." International Conference on Artificial Intelligence and Statistics. PMLR, 2025.

**Questions:**

This submission applies existing adversarial attack methods to evaluate the vulnerabilities of time series foundation models and employs established adversarial defenses to mitigate these attacks. Incorporating new insights or methodological innovations would substantially enhance the significance and contribution of the work. For example, it remains unclear what new attack designs are specifically tailored to the characteristics of time series foundation models, and what unique vulnerability patterns these models exhibit beyond those already observed in conventional time series forecasting models.

---

> ### Author Response · Authors · 2025-11-14
> **Response to Reviewer ib3J**
>
> Thank you for the careful and constructive review. We sincerely appreciate the opportunity to clarify the novelty, scope, and contributions of our work.
>
> ## Scope of the paper
>
> Our primary objective is **not** to introduce a new attack or defense algorithm. Instead, we demonstrate to the community that for these powerful TSFMs, **robustness is as critical as predictive accuracy**. Despite their rapid adoption, there has been **no standardized robustness protocol**, no agreement on how to define meaningful perturbations, and no systematic evaluation across TSFMs. This gap makes it difficult for researchers and practitioners to assess safety risks or compare models.
>
> Our contribution is therefore diagnostic rather than algorithmic: we provide a **unified, time-series–specific threat model, evaluation protocol, and analysis framework** tailored to the properties of TSFMs.
>
> ## Relation to prior work & novelty
>
> Our work is conceptually distinct from prior studies:
>
> - **[1,2]** focus on conventional supervised forecasting models (e.g., small neural nets) and propose dataset-specific attacks/defenses.
> - **[3]** studies zero-shot *LLM-based* time-series models, which differ fundamentally from pretrained TSFMs in architecture, objective, and inference pipeline.
>
> In contrast, our study is **to systematically examine adversarial robustness across single-modal, pretrained large-scale TSFMs** that operate in a zero-shot forecasting regime. These models have different inductive biases, training corpora, scaling laws, and deployment patterns from both conventional models and LLM-based variants.
>
> Because the field lacks a time-series–grounded robustness foundation, our novelty lies in:
>
> - establishing a **coherent, TS-specific threat model** (structured targets, scale-aware budgets, horizon-level constraints);
> - introducing **consistent cross-model evaluation protocols** for zero-shot TSFMs;
> - uncovering **TSFM-specific vulnerability patterns** that were not previously documented;
> - providing **practical, cross-domain defenses** aligned with real deployment constraints.
>
> We believe that in an emerging area, defining the right problem and evaluation framework is as consequential as proposing a new method.
>
> ## TSFM-Specific Insights and Vulnerability Patterns
>
> We respectfully disagree that our findings offer little new insight. The study reveals several **non-trivial, previously unreported** phenomena unique to TSFMs:
>
> - **Cross-domain adversarial fine-tuning generalizes strongly.** Fine-tuning on an unrelated dataset recovers **80–95%** of the robustness gains achieved by in-domain adversarial training (Table 3). This is unexpected in a modality with no semantic structure, and offers a practical mitigation path for zero-shot settings.
> - **Longer look-back contexts amplify vulnerability**. TSFMs exhibit a structural trade-off that architectures seek higher clean accuracy by extending context systematically increase attack susceptibility, a pattern not observed in prior TS robustness studies.
> - **Weak cross-model transfer and horizon-proximal brittleness.** These behaviors differ from CV/NLP models and point to temporal dependency mechanisms as key failure surfaces.
>
> These insights are TSFM-specific and would not be observable in studies limited to conventional or LLM-based forecasting architectures.
>
> ## Comparing TSFMs with Conventional Forecasting Models
>
> We appreciate the reviewer’s suggestion. Our goal, however, is not to determine whether TSFMs are “more” or “less” robust than conventional forecasting models. Instead, the purpose of the paper is to provide a deployment-aligned robustness diagnosis within the family of zero-shot, pretrained TSFMs.
>
> A direct comparison to conventional models would be of limited interpretive value because:
>
> - these models operate under entirely different training paradigms (supervised vs. pretrained);
> - their data requirements, inductive biases, and generalization behaviors differ substantially from TSFMs;
> - their adversarial fragility has already been thoroughly documented in prior work.
>
> Including them would broaden the paper beyond its intended scope and dilute the TSFM-specific conclusions. Our framework, however, can be directly applied to conventional models, and we will clarify this in the revision and make all code available to support such extensions.

---

> ### Comment · Reviewer_ib3J · 2025-11-14
>
> Thanks for your thoughtful feedback.
>
> Overall, this research topic is both important and timely. While time-series foundation models (TSFMs) have recently attracted significant attention, their robustness remains largely underexplored. This submission provides a comprehensive comparison, and the authors present new insights into the unique vulnerabilities of TSFMs that differ from those of conventional time-series forecasting models.
>
> Although I still believe that benchmarking TSFMs against traditional models is valuable, conducting a TSFM-focused study is still well-motivated. Adapting existing techniques to a new and timely domain, while uncovering novel observations, constitutes meaningful work, especially when supported by heavy-load and well-executed experiments.
>
> Given these strengths, I am inclined to raise my score from 4 to 6.

---

> > ### Author Response · Authors · 2025-11-14
> > **Response to Comment by Reviewer ib3J**
> >
> > Thank you very much for your encouraging feedback. We sincerely appreciate the time you have taken to review our work and are grateful that you find the study timely, well-motivated, and empirically strong.
> >
> > We fully understand and agree with your point regarding the value of benchmarking TSFMs against traditional forecasting models. Although our current focus is on establishing a TSFM-centered robustness evaluation, we recognize that comparisons with classical architectures can further contextualize robustness behaviors and strengthen the overall contribution. In the revision, we will include a dedicated comparison with conventional models using the same threat model and metrics to complement our TSFM-focused analysis.
> >
> > Thank you again for raising your score and for providing constructive suggestions.

---

> ### Author Response · Authors · 2025-11-16
> **Comparison with Conventional Models**
>
> We have added a comparison with three conventional forecasting architectures (GRU, TCN, and Informer), each trained on the target dataset using standard supervised protocols. All models are evaluated under the same PGD untargeted attack configuration using the $\text{RED}_{\text{NMAE}}$ robustness metric.
>
> The results (see Table below) reveal several key observations:
>
> - Robustness differs substantially across both model families and datasets.
> - Within conventional models, greater architectural complexity corresponds to lower robustness: Informer is noticeably more fragile than simpler architectures such as GRU and TCN. This trend is consistent with prior findings that high-dimensional and more linearized models tend to exhibit stronger adversarial vulnerability [1, 2].
> - Although TSFMs aim to provide broad, cross-domain forecasting capability, their adversarial robustness is generally weaker than that of simpler, task-specific supervised models such as TCN.
>
> We will incorporate this analysis into the revised manuscript.
>
> Table. Robustness comparison between TSFMs and conventional forecasting models.
> We report
> $RED_{NMAE}$ under PGD untargeted attacks with $\epsilon=0.5, r=0.5$. Conventional forecasting models (GRU, TCN, and Informer) are trained specifically for each dataset. Lower $\text{RED}_{\text{NMAE}}$ indicates better robustness.
>
> |  | TimsFM | Moirai | UniTS | GRU | TCN | Informer |
> | --- | --- | --- | --- | --- | --- | --- |
> | ETTh1 | 2.1038 | 0.6814 | 0.0346 | 0.1456 | 0.0510 | 0.1302 |
> | ETTh2 | 3.3649 | 0.5909 | 0.0587 | 0.1203 | 0.0043 | 0.0344 |
> | ETTm1 | 6.5620 | 1.0180 | 0.1208 | 0.1087 | 0.0522 | 0.2252 |
> | ETTm2 | 5.2638 | 1.0133 | 0.0798 | 0.1060 | 0.0023 | 0.1376 |
> | Exchange | 4.2828 | 0.5000 | 0.0271 | 0.0537 | 0.0072 | 2.0443 |
> | Weather | 5.2788 | 0.4640 | 0.1660 | 0.0422 | 0.0050 | 0.3578 |
> | Electricity | 2.1000 | 0.6019 | 0.4733 | 0.0011 | 0.2314 | 0.0212 |
>
>
> [1] Goodfellow, I. J., et al. (2015). *Explaining and harnessing adversarial examples.* ICLR.
>
> [2] Ilyas, A., et al. (2019). *Adversarial examples are not bugs, they are features.* NeurIPS.

---

### Official Review · Reviewer_WKnA · 2025-11-02

**Soundness:** 3
**Presentation:** 3
**Contribution:** 2
**Rating:** 4
**Confidence:** 3

**Summary:**

The paper systematically evaluates TSFMs under a unified adversarial framework spanning goals (untargeted/targeted), capabilities (hybrid ℓ₀/ℓ∞ budget), and knowledge (white-box/black-box), showing that even small, structured perturbations can reliably steer forecasts (e.g., flips, drifts, scaling) across domains. TSFMs show pervasive vulnerabilities (e.g., TimesFM particularly sensitive under PGD); adversarial examples often don’t transfer well across models (model-specific failure modes); points near the forecast horizon are most vulnerable; longer contexts improve clean accuracy but amplify attack impact.

**Strengths:**

1. Clear, comprehensive threat modeling & eval setup: Covers white-box (PGD) and black-box (SimBA/ZOO), targeted and untargeted goals, with unified robustness metrics across six TSFMs and eight datasets.

2. Finds pervasive but model-specific vulnerabilities and quantifies factors that modulate attack success (context length, attack location, model size).

**Weaknesses:**

1. Some robustness signals may reflect gradient obfuscation: MoE-style models appear PGD-resistant, but single-step and query-based attacks still work.

2. Technical contribution seems to be limited. I don't like to use this argument for paper review but vulnerability to adversarial attacks are well-known in the entire ML community.

**Questions:**

This is a pretty comprehensive study but the technical contribution seems to be limited. Defenses using smoothing to me is still vulnerable to adaptive attacks, and the LAT is also not new.

---

> ### Author Response · Authors · 2025-11-14
> **Response to Reviewer WKnA**
>
> Thank you for the careful and constructive review.
>
> > W1. Some robustness signals may reflect gradient obfuscation: MoE-style models appear PGD-resistant, but single-step and query-based attacks still work.
> >
>
> We fully agree with your assessment and explicitly address this precise point in the manuscript.
>
> As we state in Line 393, "...our diagnostics suggest this effect aligns with gradient obfuscation rather than genuine robustness: one-step gradient attacks and query-based black-box attacks remain effective...". To further validate this, we provided a supporting analysis using one-step gradient attacks (FGSM) in Appendix E.3 (Table 8), which, as you correctly noted, remain effective. We want to clarify that our intention was not to claim that MoE architectures provide genuine robustness, but rather to investigate these seemingly robust signals and correctly identify their cause as gradient obfuscation. We will clarify this further in the next version of the manuscript.
>
> > W2. Technical contribution seems to be limited. I don't like to use this argument for paper review but vulnerability to adversarial attacks are well-known in the entire ML community. \
> > Question: This is a pretty comprehensive study but the technical contribution seems to be limited. Defenses using smoothing to me is still vulnerable to adaptive attacks, and the LAT is also not new.
> >
>
> We agree that our work is a comprehensive study rather than a proposal for a new, singular defense mechanism.
>
> While vulnerability, in general, is known in ML, its specific manifestation, severity, and characteristics in large-scale TSFMs were unstudied. **Our goal was to demonstrate to the community that robustness is as critical as predictive accuracy for these models**, especially given their potential use in high-stakes decision-making.
>
> Therefore, our technical contribution is not a "novel" attack (we show classic attacks are alarmingly effective) or a "novel" defense. We believe that for an emerging and impactful technology like TSFMs, there is immense value in:
>
> - Identifying and demonstrating the critical, overlooked safety problem.
> - Providing a systematic framework to rigorously define and evaluate this problem.
> - Establishing a comprehensive baseline that quantifies the vulnerability across diverse, state-of-the-art models across domains.
>
> Moreover, with “not new” defenses, we uncover new and practical insights. For example, while LAT is an existing technique, we find that cross-domain LAT is remarkably effective (Table 3) for TSFMs. The discovery that fine-tuning for robustness on a completely out-of-domain dataset confers a significant portion of the robustness gains is a new, non-obvious, and practical insight for the development of future TSFMs.

---

> > ### Comment · Reviewer_WKnA · 2025-11-26
> > **Response**
> >
> > Thank you for the reply. I am at the borderline of whether to accept this (kind of) paper in general, will take a look at other reviewers' comment to make final recommendation

---

### Author Response · Authors · 2025-11-27
**Summary of Revisions**

We sincerely thank all reviewers for their thoughtful and constructive feedback. We have carefully addressed all comments and substantially revised the manuscript. Below is a summary of modifications, with each change marked in blue in the revised paper.

1. **Clarifying Novelty, Positioning, and Contribution Scope**
    - **Clarification of Contribution Boundary and Scope (Reviewers WKnA, ib3J, iwVq, 1xNi):** We have revised the *Abstract* and *Introduction* to explicitly define our work as a *diagnostic study* focused on native, non-LLM TSFMs. We have toned down generalized claims to prevent overstating novelty relative to recent LLM-based forecasting studies .
    - **Clearer Relation to Prior Work (Reviewers ib3J, iwVq):** We have expanded *Related Work* to articulate the distinction between our work and recent studies on adversarial attacks for LLM-based forecasters.
2. **Extended Experimental Analysis**
    - **Comparison with Traditional Supervised Models (Reviewers ib3J, iwVq):** To contextualize TSFM vulnerability, we have added a comprehensive comparison with supervised baselines (GRU, TCN, and Informer). These results are presented in *Section 4.1 (Table 3)* and detailed in *Appendix E.12.*
    - **Gradient Obfuscation (Reviewer WKnA):** We have updated *Section 4.1* to carefully phrase conclusions regarding model-specific failure modes.
    - **Input-space Adversarial Training Baseline (Reviewer iwVq):** To provide a fair evaluation of our proposed defense, we have implemented Input-space Adversarial Training (IAT) as a strong baseline. We have added the methodology in *Section 3.5* and *Appendix C.3*, and reported comparative results against LAT in *Section 4.3 (Table 4)* and *Appendix E.11 (Table 14)* .
3. **Writing and Visualization Improvement**
    - **Precise Variable Definitions and Metrics (Reviewer 1xNi):** We have revised *Section 3.2* and the captions of *Figure 1* and *Figure 3* to explicitly define key variables.
    - **Refined Terminology and Experimental Settings (Reviewer 1xNi):** We have corrected potentially ambiguous phrasing (e.g., removing "per step" where inaccurate) and provided clearer details on attack settings and hyperparameter choices.

We believe these revisions significantly strengthen the quality and rigor of our manuscript. We welcome any further questions or suggestions.

---

### Author Response · Authors · 2025-11-27
**Broader Motivation and Scope**

We deeply appreciate the time and effort the reviewers have dedicated to this evaluation, and we value this opportunity to articulate the broader motivation driving our research.

Robustness in time-series forecasting remains far less developed than in CV or NLP, where decades of research have established common threat models, evaluation standards, and defense practices. In contrast, the time-series community still lacks shared definitions of adversarial perturbations, agreed-upon robustness metrics, or even a clear consensus on how to assess the safety of modern pretrained forecasting models.

While existing studies have scrutinized small, supervised models, attention in the broader robustness literature remains disproportionately centered on images and language. Yet, as TSFMs rapidly emerge as general-purpose backbones for critical infrastructure, their security and reliability increasingly determine real-world decision-making outcomes.

Crucially, TSFMs differ fundamentally from CV/NLP foundation models and classical forecasting architectures in ways that make their robustness uniquely challenging. These models operate on continuous, dynamic, and often noisy signals; they must achieve zero-shot generalization across heterogeneous domains; and their specific mechanisms—such as patching, tokenization, and distributional forecasting head—introduce new, model-specific pathways for perturbation propagation. Consequently, vulnerabilities in TSFMs manifest differently from those in vision or language models and cannot simply be inferred from findings in other modalities. This **distinct combination of pretraining paradigms, architectural inductive biases, and high-stakes application contexts makes TSFM robustness an emerging problem that warrants dedicated, systematic investigation.**

We view this work as a foundational diagnostic step, an early but necessary attempt to ground the field. Our hope is that it can serve as a starting point to encourage the community to take robustness in the foundation-model era seriously. If our study helps spark broader discussion, inspires stronger methods, or motivates the development of shared standards for time-series robustness, then it will have fulfilled its purpose.

---

### Meta-Review · Area_Chair_osof · 2025-12-22

**Summary:**

This paper presents a systematic evaluation of time series foundation model regarding its adversarial robustness.

Strengths:
(1) Comprehensive threat modeling. (2) systematic empirical evaluation. (3) timely and important topic.

Weaknesses:
(1) Limited technical contribution and over-claimed novelty. (2) Lack of new insights. (3) inadequate discussion with prior, related work. (4) writing quality has room for improvement.

**Reviewer Concerns:**

The concerns regarding the related work, writing and some experimental results were addressed.

**Reviewer Scores:**

WKnA gave 4 and mentioned that this is a “borderline case”, and so is likely to increase to 6. ib3J expressed the inclination to raise from 4 to 6. The two other reviewers (2 and 4): no response, and it is unclear if they will change the scores. Overall, the paper is still likely to fall below the threshold even if both WKnA and ib3J increase their scores to 6.

---

### Decision · Program_Chairs · 2026-01-26

Reject